# Do wealth and inequality associate with health in a small-scale subsistence society?

Adrian V Jaeggi[1,2]*[†], Aaron D Blackwell[3]*[†], Christopher von Rueden[4], Benjamin C Trumble[5,6], Jonathan Stieglitz[7], Angela R Garcia[2,5,6], Thomas S Kraft[8], Bret A Beheim[9], Paul L Hooper[10,11], Hillard Kaplan[10], Michael Gurven[8]

[1]Institute of Evolutionary Medicine, University of Zurich, Zurich, Switzerland; [2]Department of Anthropology, Emory University, Atlanta, United States; [3]Department of Anthropology, Washington State University, Pulman, United States; [4]Jepson School of Leadership Studies, University of Richmond, Richmond, United States; [5]School of Human Evolution and Social Change, Arizona State University, Tempe, United States; [6]Center for Evolution and Medicine, School of Life Sciences, Arizona State University, Tempe, United States; [7]Institute for Advanced Study in Toulouse, Toulouse, France; [8]Department of Anthropology, University of California, Santa Barbara, Santa Barbara, United States; [9]Department of Human Behavior, Ecology and Culture, Max Planck Institute for Evolutionary Anthropology, Leipzig, Germany; [10]Economic Science Institute, Chapman University, Irvine, United States; [11]Department of Anthropology, University of New Mexico, Albuquerque, United States

*For correspondence:
adrian.jaeggi@iem.uzh.ch (AVJ);
aaron.blackwell@wsu.edu (ADB)

[†]These authors contributed equally to this work

Competing interest: The authors declare that no competing interests exist.

**Abstract** In high-income countries, one's relative socio-economic position and economic inequality may affect health and well-being, arguably via psychosocial stress. We tested this in a small-scale subsistence society, the Tsimane, by associating relative household wealth (n = 871) and community-level wealth inequality (n = 40, Gini = 0.15–0.53) with a range of psychological variables, stressors, and health outcomes (depressive symptoms [n = 670], social conflicts [n = 401], non-social problems [n = 398], social support [n = 399], cortisol [n = 811], body mass index [n = 9,926], blood pressure [n = 3,195], self-rated health [n = 2523], morbidities [n = 1542]) controlling for community-average wealth, age, sex, household size, community size, and distance to markets. Wealthier people largely had better outcomes while inequality associated with more respiratory disease, a leading cause of mortality. Greater inequality and lower wealth were associated with higher blood pressure. Psychosocial factors did not mediate wealth-health associations. Thus, relative socio-economic position and inequality may affect health across diverse societies, though this is likely exacerbated in high-income countries.

## Introduction

It is relatively uncontroversial that people with greater access to resources – usually operationalized as income, wealth, or broader indicators of socio-economic position, rank, or status (used interchangeably here) – are likely to be in better health as resources can be converted into better nutritional status, access to health care, or insulation against health risks. Such benefits of *absolute* rank are also commonly found in nonhuman primates (*Cowlishaw and Dunbar, 1991*; *Pusey et al., 1997*; *Snyder-Mackler et al., 2020*; *van Noordwijk and van Schaik, 1999*). However, there is increasing evidence that relative access to resources, that is, one's relative position in a socio-economic hierarchy, may also affect health. Across developed societies, there is causal evidence for a health gradient along

**eLife digest** Poverty is bad for health. People living in poverty are more likely to struggle to afford nutritious food, lack access to health care, or be overworked or stressed. This may make them susceptible to chronic diseases, contribute to faster aging, and shorten their lifespans. In high-income countries, there is growing evidence to suggest that a person's 'rank' in society also impacts their health. For example, individuals who have a lower position in the social hierarchy report worse health outcomes, regardless of their incomes. But it is unclear why living in an unequal society or having a lower social status contributes to poorer health.

One possibility is that inequalities in society are creating a stressful environment that leads to worse physical and mental outcomes. It is thought that this stress largely comes from how humans evolved to prioritize reaching a higher social status over having a long and healthy life. If this is the case, this would mean that the link between social status and health would also be present in non-industrialized communities where social hierarchies tend to be less pronounced.

To test this, Jaeggi, Blackwell et al. studied the Indigenous Tsimane population in Bolivia who live in small communities and forage and farm their own food. The income and relative wealth of 870 households from 40 Tsimane communities were compared against various outcomes, including symptoms associated with depression, stress hormone levels, blood pressure, self-rated health and several diseases.

Jaeggi, Blackwell et al. found poverty and inequality did not negatively impact all of the health outcomes measured as has been previously reported for industrialized societies. However, blood pressure was higher among people with lower incomes or those who lived in more unequal communities. But because the Tsimane people generally have low blood pressure, the differences were too small to have much effect on their health. People who lived in more unequal communities were also three times more likely to have respiratory infections, but the reason for this was unclear.

This shows that social determinants such as a person's wealth or inequality can affect health, even in communities with less rigid social hierarchies. In industrial societies the effect may be worse in part because they are compounded by lifestyle factors, such as diets rich in fat and sugar, and physical inactivity which can also increase blood pressure. This information may help policy makers reduce health disparities by addressing some of the social determinants of health and the lifestyle factors that cause them.

---

socio-economic hierarchies, independent of absolute wealth or use of health-care services (*Ecob and Davey Smith, 1999*; *Marmot et al., 1991*; *Oakes et al., 1973*; *Sorlie et al., 1995*; *Wolfson et al., 1993*). In other words, these studies find that *relative* rank – how one compares to others – is a critical variable in determining health outcomes (*Anderson et al., 2012*; *Luttmer, 2005*; *Snyder-Mackler et al., 2020*; *Wood et al., 2012*).

The steepness of socio-economic hierarchies (i.e., income or wealth *inequality*) is also associated with both physical and mental health outcomes – including self-rated health, all-cause mortality, heart disease, respiratory disease, obesity, or homicide rates – independent of absolute wealth (*Nowatzki, 2012*; *Pickett and Wilkinson, 2015*; *Wilkinson and Pickett, 2006*). While these findings are hotly debated and tests of this inequality hypothesis have been critiqued on methodological grounds (*Kondo et al., 2009*; *Lynch et al., 2004*; *Macinko et al., 2003*; *Subramanian and Kawachi, 2004*; *Wagstaff and van Doorslaer, 2000*), a formal meta-analysis on studies controlling for individual wealth found significant associations between inequality and mortality or self-rated health in high-income countries (*Kondo et al., 2009*). Thus, relative position in a socio-economic hierarchy and the steepness of such hierarchies seem to matter for health.

The most cited mechanism for such hierarchy-health associations is that hierarchies cause psycho-social stress, which in turn leads to poorer health outcomes (*Chen and Miller, 2013*; *Pickett and Wilkinson, 2015*). Chronic stress leads to altered hypothalamic-pituitary-adrenal (HPA) axis function, including chronically elevated cortisol levels. Increased cortisol can cause neural atrophy, cardiovascular damage, obesity, or immunosuppression, all resulting in increased susceptibility to chronic and infectious disease (*Aiello et al., 2018*; *García et al., 2017*; *Kunz-Ebrecht et al., 2004*; *Quon and McGrath, 2014*; *Sapolsky, 2004*). In addition, submission in status competition and learned

helplessness are associated with depression in humans and other primates (*Hagen, 2011*; *Nesse, 2000*; *Stieglitz et al., 2014*). Experimental studies in nonhuman primates show that dominance rank also affects gene expression and immune function (*Snyder-Mackler et al., 2016*; *Tung et al., 2012*). Related results in humans show that early life experiences and other forms of social stress are also associated with increases in inflammation and blunted immunological responses to cortisol (*Aiello et al., 2018*; *Miller et al., 2014*; *Miller et al., 2011*; *Miller et al., 2009*).

But why are hierarchies stressful or otherwise detrimental to health? An evolutionary-medicine perspective suggests that many detrimental health outcomes may result from adaptive *tradeoffs*, as fitness gains are prioritized over detrimental health outcomes, *developmental constraints*, as long-term negative health effects result from short-term accommodations to conditions during development, or from evolutionary *mismatch*, as our bodies struggle to deal with conditions atypical for our species (*Eaton et al., 2002*; *Gluckman et al., 2016*; *Lea et al., 2017*; *Nesse and Williams, 1994*; *Wells et al., 2017*). Given the consistent fitness benefits of high status (*Stulp et al., 2016*; *Von Rueden and Jaeggi, 2016*), and given that fitness is always relative, humans arguably have evolved motivations for status-striving that are independent of one's absolute access to resources (*Johnson et al., 2012*; *Shenk et al., 2016*). Status-striving activates the stress response, and not just for low-rankers: depending on how rank is achieved and maintained, high- or low-ranking individuals may be more stressed (*Abbott et al., 2003*; *Sapolsky, 2005*). Crucial to who is stressed is the availability of social support, which can be as or even more important for health and fitness as rank per se (*Sapolsky, 2005*; *Snyder-Mackler et al., 2020*). Other factors primarily impact low-ranking individuals: in many primate (and some human) societies, subordinates are regularly subjected to aggression and intimidation by higher-ranking individuals (*Silk, 2003*), resulting in the lack of control and learned helplessness that often cause depression (*Sapolsky, 2005*; *Sapolsky, 2004*).

Greater inequality, that is, steeper hierarchies, entails more skewed payoff distributions and thus also favors more intense competition and risk-taking as behavioral strategies, especially among low-ranking individuals; this is argued to explain the persistent association between income inequality and homicide rates as most homicides result from escalated contests over status (*Daly, 2016*; *Daly and Wilson, 1997*). If skewed payoff distributions and oppression of low-rankers favor life-history strategies focused on short-term payoffs (i.e., 'faster' life-history strategies sensu *Wells et al., 2017*), this could also explain hierarchy-health associations via present-oriented decision-making at the expense of long-term health (*Daly and Wilson, 1997*; *Griskevicius et al., 2011*; *Pepper and Nettle, 2014*). These relationships are expected even when hierarchies are based on prestige, rather than dominance, since prestige-based hierarchies still correlate with social support, insulation against shocks, influence, sense of control, and access to mates (*Gurven et al., 2000*; *Sugiyama and Sugiyama, 2003*; *von Rueden et al., 2014*).

Thus, stress and negative health consequences due to socio-economic hierarchies can result from perpetual status-striving, unequal distribution of social support, lack of control and learned helplessness, intensified competition especially among low-ranking individuals, and from physiological accommodations to generally 'faster' life histories. In short, hierarchies may cause stress and affect health largely because individuals engage in competitive strategies that function to maximize fitness at the expense of health, while failure to succeed in such competition negatively affects mental health. In addition, if hierarchies constrain access to resources individuals may face developmental constraints, causing long-term tradeoffs that negatively impact health.

While such adjustments of physiology and behavior to the local competitive environment may in principle generalize to all human societies, the effects of hierarchy on health may be exacerbated in industrialized, high-income countries due to mismatch. Specifically, such societies could represent a mismatch with the ancestral environments in which our competitive strategies have evolved because (i) socio-economic hierarchies may be steeper and more rigid than was typical of our hunter-gatherer ancestors (*Borgerhoff Mulder et al., 2009*; *Kaplan et al., 2009*), and include features such as lack of kin support, limited upward mobility, structural violence, and systemic racism, all of which are well-known to negatively affect health (*Gravlee, 2009*; *Sapolsky, 2004*); and (ii) novel lifestyle factors such as obesogenic diets, lack of physical activity, and chronic inflammation turn previously relatively harmless responses, such as temporarily elevated blood pressure or depressed mood, into 'mismatch' diseases, such as hypertension, atherosclerosis, and major depression (*Gurven et al., 2012*; *Kaplan et al., 2017*; *Miller and Raison, 2016*; *Stieglitz et al., 2015*). In sum, mismatch diseases often arise

when risk factors that used to elicit an adequate acute response become chronic problems, which could well be the case with modern socio-economic hierarchies, and interact with novel lifestyle factors that push our physiology into novel and unhealthy ranges.

In summary, humans, much like other primates, are sensitive to their relative rank and the distribution of fitness outcomes, and adjust behavior and physiological responses accordingly, resulting in negative influences of hierarchy on health. Several open questions remain, however. First, the inequality hypothesis remains hotly debated since parsing inequality from other correlated variables is difficult and requires careful statistical methods. Second, it remains unclear to what extent the observed health consequences of relative status and inequality in high-income countries (i) represent tradeoffs of potentially adaptive responses to lower relative rank and/or to inequality, or (ii) are caused by evolutionary mismatch, that is, novel conditions that cause maladaptive outcomes. If health consequences stem from tradeoffs from adaptive responses, then hierarchy should be associated with health in any population, independent of absolute access to resources. However, if the impacts of status and inequality are caused by evolutionary mismatch, then we would not expect detrimental effects on health in all societies, though we might observe related physiological responses in a subclinical range.

Small-scale societies practicing traditional subsistence lifestyles (henceforth 'subsistence societies') are an important test case for the universality of hierarchy-health associations as they generally have more informal, egalitarian hierarchies with relatively high individual autonomy and mobility (*Borgerhoff Mulder et al., 2009*; *Kaplan et al., 2009*; *Mattison et al., 2016*), and suffer from infectious rather than chronic disease as major sources of morbidity and mortality (*Eaton et al., 1988*; *Gurven et al., 2007*; *Gurven et al., 2016*; *Gurven and Kaplan, 2007*; *Kaplan et al., 2017*; *Pontzer et al., 2018*). Further, individuals in many subsistence societies have immune systems that are well calibrated by frequent exposure to pathogens and microbiota, and predominantly experience acute responses to infections (*Blackwell et al., 2016a*; *McDade, 2005*), unlike the chronic low-grade inflammation that links stress to hypertension, cardiovascular disease, and depression in high-income countries (*Gurven et al., 2008a*). Lastly, competition for mates and resources in such societies is usually fairly local, meaning that the scale at which relative rank and inequality should be measured is more recognizable than in large-scale modern societies with mass media, where people are simultaneously part of many hierarchies. Thus, subsistence societies may help us discern whether associations between hierarchy and health are caused by tradeoffs expected in any society, by evolutionary mismatch in modern, industrialized populations, or a combination of both.

Few studies have examined associations between rank or inequality and health in subsistence societies. Among Dominican farmers, socio-economic indicators were unrelated to cortisol levels whereas local influence was associated with lower cortisol (*Decker, 2000*). Among egalitarian Garisakang horticulturalists in Papua New Guinea, higher income coming from greater market exposure was associated with higher cortisol, whereas other locally relevant measures of wealth and status were not (*Konečná and Urlacher, 2017*). While results are mixed, there is some converging evidence that suggests market integration generates psychosocial stress in subsistence societies, arguably due to the threat of cultural loss and discrimination often experienced through contact with majority groups.

Among Tsimane forager-horticulturalists in Bolivia, it has been reported that traditional forms of status generally support a status-health gradient, but studies on income or wealth show mixed results. In a sample of four communities, politically influential men had lower cortisol and a lower incidence of respiratory infection, though there were also many null results, and higher income was associated with higher cortisol (*von Rueden et al., 2014*). In one village, women's political influence was associated with improved growth and health outcomes for their children (*Alami et al., 2020*). Across 13 Tsimane villages, relative wealth was associated with better self-reported health (*Undurraga et al., 2010*); however, average self-reported health was lower in wealthier villages. In a larger sample of villages, relative income associated with lower body mass index (BMI) among individuals with smaller support networks (*Brabec et al., 2007*).

In terms of the relationship between inequality and health within communities, studies among Tsimane have also shown mixed results. One study found no association between income inequality and body fat (*Godoy et al., 2005*), but income inequality was associated with more negative emotions (*Godoy et al., 2006*). Greater wealth inequality did not associate with self-reported health in one study (*Undurraga et al., 2010*) but did associate with better self-reported health and lower self-reported

stress in another, controlling for individual and village wealth level (*Undurraga et al., 2016*). Overall, these results provide mixed evidence for associations between inequality and health.

Here, we test for links between hierarchy and health among the Tsimane, expanding upon previous studies in several ways. First, we simultaneously assess the effects of within-community-relative wealth, mean community wealth, and community-level wealth inequality. Second, while previous studies have mostly relied on just one or two indirect health outcomes such as BMI, we include 13 different dependent variables (*Table 1*) capturing various health outcomes, including infectious disease morbidity, psychological well-being, social conflicts and connections, and other stressors. Third, we explicitly test whether these psychological and social variables and other stressors (henceforth 'psychosocial variables') mediate links between wealth and health as predicted if the adverse health effects of hierarchy occur through psychosocial stress. Note though that some of these 'psychosocial' variables may also be associated with health through more direct mechanisms, for example, non-social problems (food insecurity, debt, etc.) may cause stress but also represent poorer access to resources, which could affect health through energetic constraints. Fourth, we greatly increase the sample size relative to previous studies with inequality measured in 40 communities and wealth in 871 households, representing approximately one quarter of the Tsimane population (see *Table 1*, *Figure 1—figure supplement 1*). Thus, our study represents the most comprehensive test of hierarchy-health associations in a subsistence society.

We specifically test the following predictions stemming from the hypotheses that relative socio-economic position as well as the steepness of socio-economic hierarchies affect health and well-being, and that these effects are mediated by psychosocial stress.

> P1: Higher relative wealth is associated with better psychosocial and health outcomes.
> P2a: Greater wealth inequality is associated with worse psychosocial and health outcomes, and
> P2b: this should hold especially for low-rankers.
> P3: Psychosocial variables mediate wealth and inequality-health links found under P1 and P2.

*Table 1* gives an overview of all variables used to test these predictions.

## Study population

The Tsimane are a population of >16,000 Indigenous Amerindians living in >90 communities at the edge of the Amazon basin in lowland Bolivia. Tsimane communities consist of dispersed household clusters tied together by networks of kinship, cooperative production and consumption (*Hooper et al., 2015*; *Jaeggi et al., 2016*), as well as usually a school and soccer field. Community meetings convene to discuss and resolve important matters, including conflicts within the community. As such, we treat the community as the salient scale of status competition (*Alami et al., 2020*; *von Rueden et al., 2018*; *von Rueden et al., 2008*; *von Rueden et al., 2019*; *von Rueden et al., 2014*) and calculated relative wealth and inequality at this level.

The Tsimane remained relatively isolated from the larger Bolivian economy until the 1970s and still widely practice traditional subsistence (swidden horticulture, hunting, and fishing), which contributes >90% of their calories (*Gurven et al., 2017*; *Kraft et al., 2018*). Cattle, introduced by missionaries and ranchers, are owned by a small minority of Tsimane. Over the past few decades, wage labor opportunities with loggers or ranchers and produce sales in the local market towns of San Borja and Yucumo have been increasing, as have formal schooling, Spanish fluency, and access to modern amenities such as electricity and health care. The population thus exhibits quantifiable gradients of modernization (see *Figure 1*).

In terms of morbidity and mortality, the Tsimane are characterized by high infectious disease burden, with respiratory infections as the leading cause of death at all ages (*Gurven et al., 2007*). Additionally, parasites such as helminths and giardia are highly prevalent (*Blackwell et al., 2013*; see also *Table 2*). These conditions result in frequent, acute immune responses (*Blackwell et al., 2016a*) but still a low incidence of chronic conditions such as hypertension or atherosclerosis, due to high levels of physical activity and other protective factors (*Gurven et al., 2012*; *Gurven et al., 2016*; *Gurven et al., 2009*; *Kaplan et al., 2017*).

**Table 1.** Overview of study variables and descriptive statistics.

For an overview of the sample relative to all people known to the Tsimane Health and Life History Project and at risk of having wealth data, see Figure 1—figure supplement 1.

| Variable | N | Obs | Median | SD | Min | Max |
|---|---|---|---|---|---|---|
| **Adult outcomes: psychosocial** | | | | | | |
| Depression (possible range 16–64) | 528 | 670 | 40.0 | 7.1 | 23.0 | 62.0 |
| Conflicts (possible range 0–4) | 342 | 401 | 2.0 | 0.7 | 0.0 | 4.0 |
| Labor partners (count)* | 304 | 399 | 2.0 | 2.0 | 1.0 | 13.0 |
| Non-social problems (possible range 0–7) | 339 | 398 | 3.0 | 1.2 | 0.0 | 7.0 |
| Urinary cortisol (pg/ml) | 588 | 811 | 155,191 | 149,602 | 93 | 851,308 |
| **Adult outcomes: health** | | | | | | |
| Body mass index (kg/m$^2$) † | 1901 | 5179 | 23.3 | 2.8 | 16.0 | 36.6 |
| Systolic blood pressure (mmHg) | 1622 | 3195 | 110.0 | 12.8 | 60.0 | 190.0 |
| Diastolic blood pressure (mmHg) | 1622 | 3195 | 70.0 | 10.0 | 24.0 | 136.0 |
| Self-rated health (1 excellent to 5 very bad) | 1307 | 2523 | 4.0 | 0.5 | 1.0 | 5.0 |
| Total morbidity (possible range 0–18)‡ | 1306 | 1542 | 2.0 | 1.1 | 0.0 | 5.0 |
| Infections/parasites (yes/no) ‡ | 1306 | 1542 | 25.2% | | | |
| Respiratory disease (yes/no) ‡ | 1306 | 1542 | 21.9% | | | |
| Gastrointestinal (yes/no) ‡ | 1306 | 1542 | 36.3% | | | |
| **Adult predictors** | | | | | | |
| Age (years) | 1931 | 5383 | 35.0 | 15.1 | 16.0 | 91.0 |
| Sex (0 = female, 1 = male) | 1931 | 5383 | 46.2 | | | |
| **Juvenile outcomes: health** | | | | | | |
| Body mass index (kg/m$^2$) † | 1765 | 4747 | 16.6 | 2.1 | 10.2 | 27.6 |
| Total morbidity (count) ‡ | 1423 | 1569 | 1.0 | 0.8 | 0.0 | 4.0 |
| Infections/parasites (yes/no) ‡ | 1423 | 1569 | 13.6% | | | |
| Respiratory disease (yes/no) ‡ | 1423 | 1569 | 42.4% | | | |
| Gastrointestinal (yes/no) ‡ | 1423 | 1569 | 41.2% | | | |
| **Juvenile predictors** | | | | | | |
| Age (years) | 1772 | 4783 | 7.0 | 4.1 | 0.0 | 15.0 |
| Sex (0 = female, 1 = male) | 1772 | 4783 | 49.6 | | | |
| **Household predictors** | | | | | | |
| Household size | 871 | 1045 | 4.0 | 2.7 | 1.0 | 14.0 |
| Household wealth (Bs) | 871 | 1045 | 7675 | 5675 | 386 | 56,664 |
| **Community predictors** | | | | | | |
| Community size (adults > 15) | 40 | 55 | 72.0 | 81.2 | 27.0 | 346.0 |
| Distance to market town (km) | 40 | 55 | 43.0 | 44.2 | 5.0 | 140.0 |
| Mean community wealth (Bs) | 40 | 55 | 8373 | 2331 | 3930 | 16,250 |
| Community wealth inequality (Gini) | 40 | 55 | 0.27 | 0.07 | 0.15 | 0.53 |

*Reverse coded in analyses to make higher values worse outcomes.

†Whether higher or lower body mass index is better is a bit ambiguous: in high-income countries higher body mass index is associated with worse health, lower status, and greater inequality, whereas in low-income countries the reverse may be true.

‡See Table 2 for an overview of the most common morbidities by category.

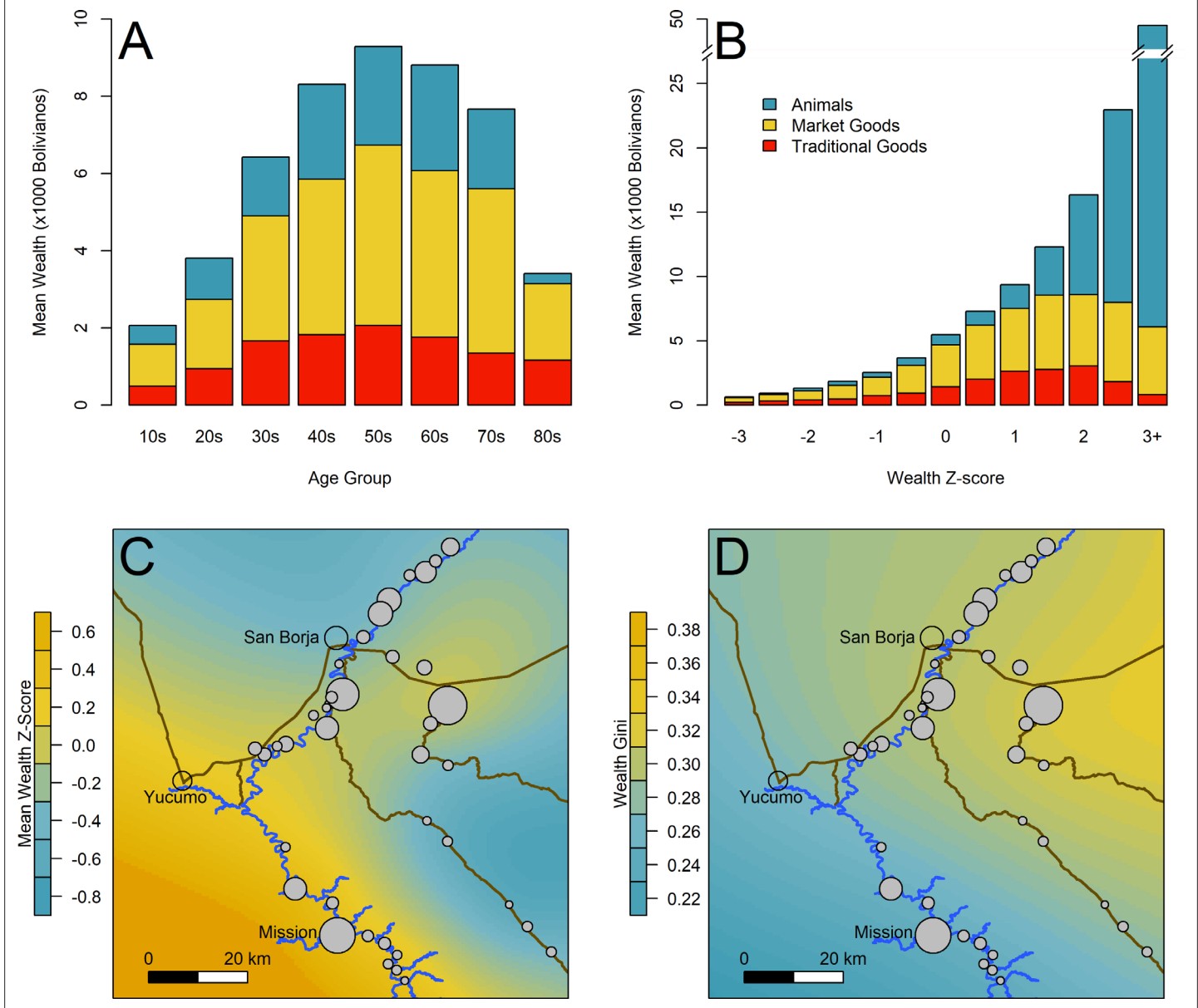

**Figure 1.** Overview of wealth and inequality distributions. (**A**) Mean wealth by age of household head. (**B**) Mean wealth by population-level wealth Z-score. (**C**) Map of study communities (n = 40) and mean wealth at the community level. (**D**) Map of community-level wealth inequality. Note: (**A**) and (**B**) use raw wealth, while (**C**) and (**D**) are based on age-corrected values. Heat maps in (**C**) and (**D**) give a rough sense of the distribution; circle size indicates the number of sampled households (range = 9–81). Data for individual villages are not directly shown to protect confidentiality. Yucumo and San Borja are local market towns inhabited by non-Tsimane, Mission is the site of a Catholic mission and the largest Tsimane settlement.

The online version of this article includes the following figure supplement(s) for figure 1:

**Figure supplement 1.** Overview of the sample.

## Results

### Tsimane wealth and inequality

Because wealth varied considerably by age (*Figure 1A*), we used an age-corrected measure of relative wealth that reflects one's wealth relative to this age trajectory (see Materials and methods). This corrects for random variation in the age structure of sampled communities and arguably better captures the essence of relative socio-economic rank: what matters is how one compares to others, relative to general trends such as wealth (status, influence, etc.) accumulating with age. At the high end

**Table 2.** Overview of the most common morbidities.

Three of the most common Clinical Classifications System (CCS) categories (number in parentheses) and the six most prevalent diagnoses within each category (in decreasing order down rows, ICD-10 codes in parentheses). Musculoskeletal conditions (CCS 13) were also common but not analyzed independently.

| Infectious and parasitic diseases (CCS 1) | Diseases of the respiratory system (CCS 8) | Diseases of the digestive system (CCS 9) |
|---|---|---|
| Pediculosis due to *Pediculus humanus capitis* (B85.0) | Acute nasopharyngitis (common cold) (J00) | Intestinal helminthiasis (B82.0) |
| Tinea unguium (B35.1) | Acute streptococcal tonsillitis; unspecified (J03.00) | Infectious gastroenteritis and colitis (A09) |
| Candidiasis of vulva and vagina (B37.3) | *Streptococcal pharyngitis* (J02.0) | Dyspepsia (K30) |
| Pediculosis; unspecified (B85.2) | Acute upper respiratory infection; unspecified (J06.9) | Gastro-esophageal reflux disease with esophagitis (K21.0) |
| Superficial mycosis; unspecified (B36.9) | Acute bronchitis due to *Mycoplasma pneumonia* (J20.0) | Giardiasis (lambliasis) (A07.1) |
| Necatoriasis (B76.1) | Bronchopneumonia; unspecified organism (J18.0) | Gastritis; unspecified, without bleeding (K29.70) |

of the wealth distribution (*Figure 1B*), much of the variation was driven by livestock, especially cattle. *Figure 1C–D* illustrates variation in mean wealth and wealth inequality among the study communities. Mean wealth was generally lowest in communities located in the interior forest (*Figure 1C*, bottom right), which are remote and inaccessible by road for much of the year (due to washed-out bridges); and in those communities downriver from San Borja (*Figure 1C*, top), which experience frequent flooding and are within or adjacent to a protected bioreserve that limits resource extraction. Somewhat unexpectedly, mean wealth was higher further from the market town of San Borja (correlation between mean wealth and distance to market $r = 0.36$, df = 38, p=0.02). We operationalized inequality by calculating community-level Gini coefficients for age-corrected wealth (see Materials and methods). Wealth inequality was generally higher in communities closer to the market towns of San Borja and Yucumo, where Tsimane can sell produce and purchase market goods, though some villages near towns also show low inequality (*Figure 1D*) (correlation between Gini and distance to market $r = –0.38$, df = 38, p=0.01). Inequality was marginally lower in richer communities ($r = –0.22$, df = 38, p=0.17). Community size was not significantly related to distance ($r = –0.18$, p=0.26), mean wealth ($r = 0.11$, p=0.50), or inequality ($r = 0.00$, p=0.99). In sum, villages near towns had both higher inequality and lower mean wealth due to both more wealthy individuals and more very poor individuals in these communities.

## Modeling strategy

To examine the effects of household wealth and community wealth inequality on psychosocial or health outcomes, we used Bayesian multilevel models with appropriate controls and random effects at the individual, household, and community level (see Materials and methods). Wealth was divided into relative wealth, centered on the community mean, and mean community wealth. Operationalizing wealth this way means we are in principle able to tease apart within-community wealth differentials, that is, one's position in the local socio-economic hierarchy, from community-level differences in access to resources, that is, mean community wealth (*Kreft et al., 1995*). However, in practice, models with wealth centered on the village produced virtually identical estimates to models with wealth centered on the sample as a whole (see *Supplementary file 1a-1m*), largely because villages did not differ strongly in mean wealth (median –0.03, range: –1.0–0.66 Z-scores, 80% between –0.43 and 0.37). Thus, community-relative and population-relative wealth were highly correlated ($r = 0.92$).

Bayesian models produce a posterior distribution of parameter estimates that can be summarized in various ways (*McElreath, 2020*). Here, we provide coefficient plots (*Figures 2–5*) showing posterior medians, as well as 75% and 95% highest posterior density intervals; we also provide prediction plots as supplements to these figures. In the text, we report results as standardized coefficients (β) for

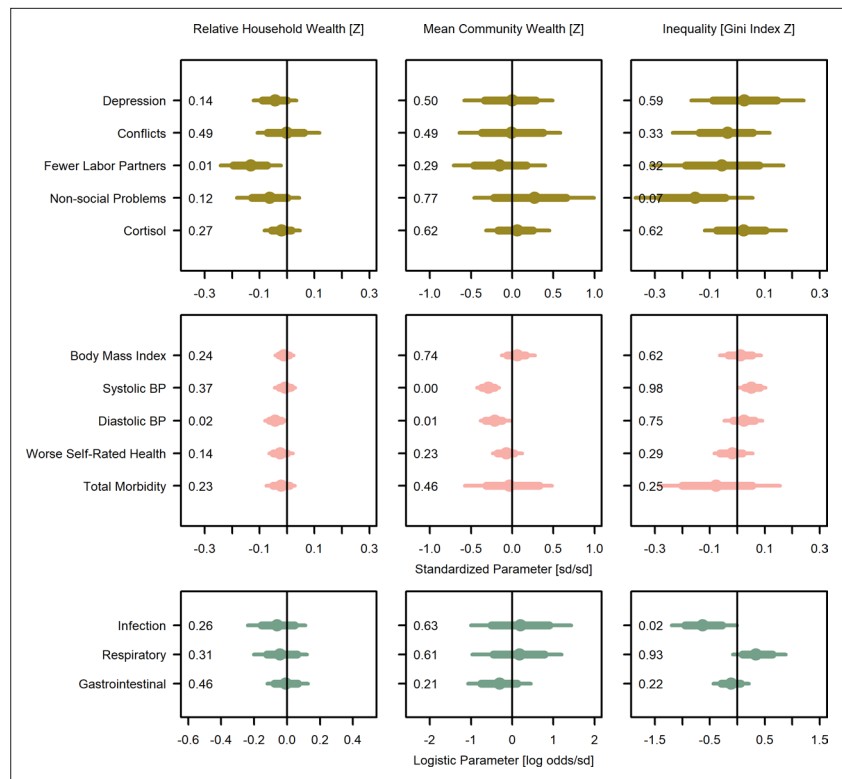

**Figure 2.** Wealth and inequality posterior parameter values for models with adults (>15 years). Points are posterior medians and lines are 75% (thick) and 95% (thin) highest posterior density intervals. Numbers in each panel represent the proportion of the posterior distribution that is greater than zero ($P_{>0}$). All models control for age, sex, distance to market town, and community size. Rough categories of dependent variables (psychosocial, continuous health outcomes, and binary health outcomes) are distinguished by rows and colors. For the first two rows, the outcomes are expressed as Z-scores, the bottom row as log odds. See *Figure 2—figure supplement 1*, *Figure 2—figure supplement 2*, and *Figure 2—figure supplement 3* for predicted associations of household wealth, community wealth, and wealth inequality, respectively.

The online version of this article includes the following figure supplement(s) for figure 2:

**Figure supplement 1.** Predicted conditional effects of relative household wealth on all psychosocial and health outcomes for adults.

**Figure supplement 2.** Predicted conditional effects of mean community wealth on all psychosocial and health outcomes for adults.

**Figure supplement 3.** Predicted conditional effects of wealth inequality (Gini coefficients) on all psychosocial and health outcomes for adults.

Gaussian models or as log odds (β) and odds ratios (ORs) for logistic models, both represented by the posterior mean, as well as the proportion of the posterior above zero ($P_{>0}$), that is, the likelihood of a positive association. Higher or lower values of this number represent stronger certainty for a non-zero effect, while values near 0.5 indicate complete uncertainty about the direction of an association, if any. In addition, we report Cohen's *d* as a standardized measure of effect size to allow comparison between continuous and binary variables; *d* is reported as the posterior median and the median absolute deviation (MAD) (a more robust measure of dispersion than the standard deviation). For simplicity, we refer to effect sizes of *d* > 0.2 as 'strong,' those >0.1 as 'moderate,' and consider the rest to be 'weak' though potentially still suggestive of a general pattern. Similarly, we refer to posterior support of >0.975 (or <0.025, if negative) as 'high certainty' and those with support >0.875/<0.125 as 'moderate certainty,' corresponding to the entire 95% or 75% highest posterior density intervals respectively not overlapping with 0, and we consider the rest to be 'uncertain.' However, we encourage readers to use the full information on the posteriors to inform their own inference. Means and 95% credible intervals for all parameters are reported in *Supplementary file 1a-1o*. These tables also provide Bayesian $R^2$

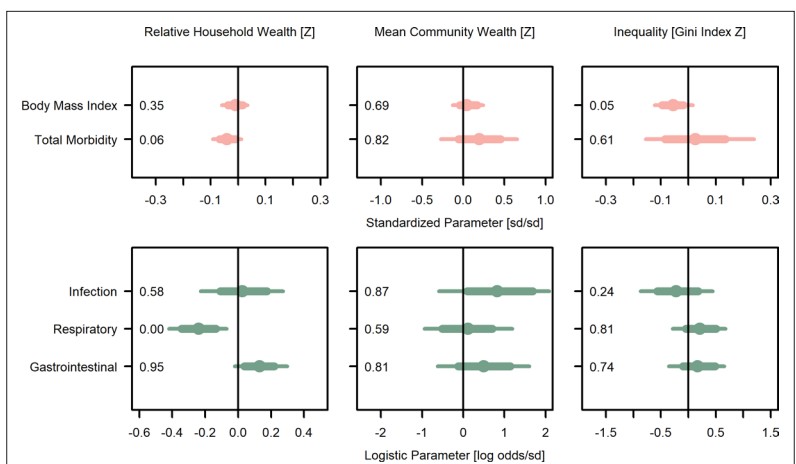

**Figure 3.** Wealth and inequality posterior parameter values for models with juveniles (≤15 years). Points are posterior medians and lines are 75% (thick) and 95% (thin) highest posterior density intervals. Numbers in each panel represent the proportion of the posterior distribution that is greater than zero ($P_{>0}$). All models control for age, sex, distance to market town, and community size. Rough categories of dependent variables (continuous health outcomes and binary health outcomes) are distinguished by rows and colors. For the first row, the outcomes are measured as Z-scores, the bottom row as log odds. See *Figure 3—figure supplement 1* for predicted associations of household wealth, community wealth, and wealth inequality.

The online version of this article includes the following figure supplement(s) for figure 3:

**Figure supplement 1.** Predicted conditional effects of household wealth, community wealth, and inequality (Gini coefficients) on all health outcomes for juveniles (<15 years).

(*Gelman et al., 2019*) as a goodness-of-fit measure, indicating that in most models the predictors and random effects jointly explained about 20–40% of the variance in the data ($R^2$ range: 0.16–0.91).

## Is wealth related to health outcomes?

Overall, for adults, household wealth was associated, with various effect sizes and degrees of confidence, with beneficial health outcomes except gastrointestinal illness, which showed no association (*Figure 2*; *Supplementary file 1f-1m*). Community mean wealth had more mixed associations with health outcomes. Specifically, household wealth was associated with lower systolic blood pressure ($\beta$ = −0.01, $P_{>0}$=0.37, Cohen's $d$ = −0.01 [0.02]) and lower diastolic blood pressure ($\beta$ = −0.04, $P_{>0}$=0.02, $d$ = −0.05 [0.02]), though both effect sizes were small and only the latter association had high certainty. Community mean wealth was strongly and with high certainty associated with lower systolic ($\beta$ = −0.29, $P_{>0}$=0.00, $d$ = −0.34 [0.09]) and diastolic ($\beta$ = −0.21, $P_{>0}$=0.01, $d$ = −0.24 [0.11]) blood pressure. Household wealth also associated with better self-rated health (reverse coded $\beta$ = −0.02, $P_{>0}$=0.14, $d$ = −0.03 [0.02]), lower odds of infectious ($\beta$ = −0.06, $P_{>0}$=0.26, $d$ = −0.02 [0.05], OR = 0.94) and respiratory ($\beta$ = −0.04, $P_{>0}$=0.69, $d$ = −0.03 [0.05], OR = 0.96) illness, and lower total morbidity ($\beta$ = −0.02, $P_{>0}$=0.23, $d$ = −0.02 [0.04]), though again most effect sizes were small and there was high uncertainty. There was no evidence for an association with gastrointestinal infection. However, there was a moderate though uncertain association between community mean wealth and lower gastrointestinal illnesses ($\beta$ = −0.32, $P_{>0}$=0.21, $d$ = −0.16 [0.21], OR = 0.72). Household wealth was weakly and uncertainly associated with lower BMI ($\beta$ = −0.01, $P_{>0}$=0.24, $d$ = −0.02 [0.04]), but community mean wealth was weakly associated with higher BMI ($\beta$ = 0.06, $P_{>0}$=0.74, $d$ = 0.12 [0.20]). Using population-relative wealth, rather than community-relative wealth had little effect on these associations (*Supplementary file 1a-1m*). In sum, despite mostly small effect sizes and high uncertainty, the general pattern was for wealthier adults to have better outcomes.

For juveniles ≤ 15 years of age (*Figure 3*; *Supplementary file 1n,o*), household wealth was weakly associated with lower total morbidity ($\beta$ = −0.04, $P_{>0}$=0.06, $d$ = −0.06 [0.04]), and in particular, moderately lower odds of respiratory illness ($\beta$ = −0.24, $P_{>0}$<0.01, $d$ = −0.13 [0.05], OR = 0.79). However, both household and community mean wealth were associated with higher odds of gastrointestinal illness ($\beta$ = 0.13, $P_{>0}$=0.95, $d$ = 0.07 [0.04], OR = 1.14; $\beta$ = 0.49, $P_{>0}$=0.81, $d$ = 0.27 [0.30], OR = 1.63)

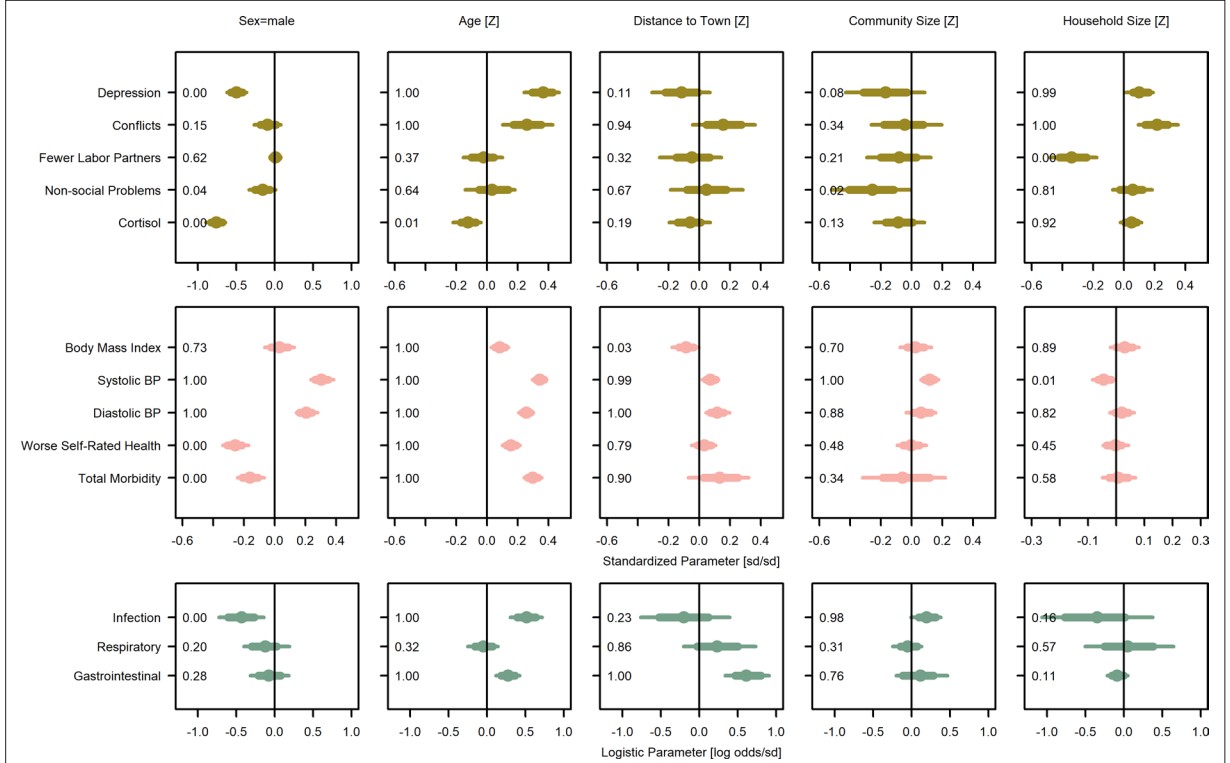

**Figure 4.** Covariate posterior parameter values for models with adults (>15 years). Points are posterior medians and lines are 75% (thick) and 95% (thin) highest posterior density intervals. Numbers in each panel represent the proportion of the posterior distribution that is greater than zero (P>0). Full models are given in *Supplementary file 1a-1m*. Rough categories of dependent variables (psychosocial, continuous health outcomes, and binary health outcomes) are distinguished by rows and colors. For the first two rows, the outcomes are measured as Z-scores, the bottom row as log odds.

and community mean wealth was associated with other infections ($\beta$ = 0.81, P>0=0.87, *d* = 0.44 [0.37], OR = 2.25) and higher total morbidity ($\beta$ = 0.05, P>0=0.82, *d* = 0.31 [0.33]) with mostly strong effect sizes but high uncertainty. In sum, for juveniles, wealth was moderately associated with reduced risk of respiratory illness, while community wealth was strongly associated with several negative health outcomes.

## Is inequality related to health outcomes?

For adults, inequality was associated with higher levels of three morbidity-related outcomes and lower levels of four outcomes (*Figure 2*; *Supplementary file 1f,m*). Consistent with predictions of worse health with inequality (P2a), greater inequality was weakly associated with higher blood pressure (systolic: $\beta$ = 0.05, P>0=0.98, *d* = 0.06 [0.03]; diastolic: $\beta$ = 0.02, P>0=0.75, *d* = 0.03 [0.04]), and strongly with a greater likelihood of respiratory illness ($\beta$ = 0.35, P>0=0.93, *d* = 0.20 [0.13], OR = 1.36). Despite these harmful associations with inequality, people in more unequal communities had a strongly lower likelihood of other infections ($\beta$ = −0.62, P>0=0.02, *d* = −0.33 [0.16], OR = 0.54) and to a more uncertain degree, total morbidity ($\beta$ = −0.07, P>0=0.25, *d* = −0.07 [0.13]), and gastrointestinal infections ($\beta$ = −0.12, P>0=0.22, *d* = −0.06 [0.09], OR = 0.89). Associations with BMI were negligible ($\beta$ = 0.01, P>0=0.62, *d* = 0.03 [0.08]).

In contrast, for juveniles (*Figure 3*; *Supplementary file 1m,o*), BMI was lower in more unequal communities ($\beta$ = −0.06, P>0=0.05, *d* = −0.08 [0.05]). Inequality had little effect on total morbidity and was moderately associated with less infectious illness ($\beta$ = −0.23, P>0=0.24, *d* = −0.13 [0.19], OR = 0.79), but greater respiratory illness ($\beta$ = 0.21, P>0=0.81, *d* = 0.11 [0.13], OR = 1.23) and gastrointestinal illness ($\beta$ = 0.17, P>0=0.74, *d* = 0.10 [0.13], OR = 1.19), both of which are highly prevalent among juveniles.

## Is wealth related to psychosocial outcomes?

For adults, greater household wealth was associated with better outcomes in four of five psychological and social measures, with no association for the fifth (*Figure 2*; *Supplementary file 1a-1e*). Household wealth was strongly and with high certainty associated with having more labor partners (reverse coded $\beta$ = –0.13, $P_{>0}$=0.01, $d$ = –0.49 [0.20]), and weakly and uncertainly, with fewer depressive symptoms ($\beta$ = –0.04, $P_{>0}$=0.14, $d$ = –0.05 [0.05]), fewer non-social problems (i.e., self-reported concerns over food insecurity, debt, and illness; $\beta$ = –0.06, $P_{>0}$=0.12, $d$ = –0.08 [0.07]), and lower urinary cortisol ($\beta$ = –0.02, $P_{>0}$=0.27, $d$ = –0.02 [0.04]). There was no support for an association with social conflicts. Unlike household wealth, community mean wealth was not clearly associated with any psychosocial outcome, though there were strong but uncertain associations with more labor partners (reverse coded $\beta$ = –0.16, $P_{>0}$=0.29, $d$ = –0.77 [0.94]) but also more non-social problems ($\beta$ = 0.28, $P_{>0}$=0.77, $d$ = 0.33 [0.45]).

## Is inequality related to psychosocial outcomes?

Contrary to predictions, inequality was largely associated with fewer stressors and psychological or social problems (*Figure 2*; *Supplementary file 1a-1e*). The strongest evidence was for fewer non-social problems in more unequal communities ($\beta$ = –0.15, $P_{>0}$=0.07, $d$ = –0.17 [0.13]), with weak evidence for fewer conflicts ($\beta$ = –0.04, $P_{>0}$=0.33, $d$ = –0.01 [0.09]), and more labor partners ($\beta$ = –0.05, $P_{>0}$=0.32, $d$ = –0.29 [0.38]) with more inequality.

## Do psychosocial variables mediate relationships between wealth or wealth inequality and health?

We tested the prediction (P3) that the effects of wealth or inequality on health were mediated via psychosocial pathways using formal mediation analysis (*Baron and Kenny, 1986*; *MacKinnon et al., 2007*). Specifically, this involves estimating the association between wealth/inequality and psychosocial variables ('path *a*'), as well as between psychosocial variables and health outcomes ('path *b*'); if both are statistically significant and the association between wealth/inequality and health outcomes ('path *c*', or *direct effect*) is weaker, then there is evidence that there is an *indirect effect* of wealth/inequality on health via psychosocial variables (i.e., the psychosocial variable is a mediator). As reported above, paths *a* were mostly supported for household wealth, that is, household wealth was associated with four of the five psychosocial variables, but not for community wealth or inequality. *Supplementary file 1q-1s* presents mediation analyses with each health outcome variable and each psychosocial variable as a potential mediator, including estimates of the direct (path *c*, as reported above) and indirect effects, the mediator effects (path *b*), and the proportion mediated (indirect effect/total effect). See Appendix 1 for a discussion and graphical depiction of the causal relationships assumed by this mediation approach.

The only convincing evidence for mediation was found for depression and non-social problems mediating the effect of household wealth on diastolic blood pressure; specifically, household wealth was negatively associated with diastolic blood pressure (path *c*) as well as with depression and non-social problems (paths *a*; see above, *Figure 2*), and both depression ($\beta$ = –0.03, $P_{>0}$=0.20) and non-social problems ($\beta$ = –0.08, $P_{>0}$=0.05) were themselves negatively associated with diastolic blood pressure (paths *b*). However, there were no other cases where both paths *a* and *b* were well supported, the indirect effects of household wealth, community wealth, or inequality were virtually always zero for any mediator (including depression and non-social problems), and the proportion mediated was generally small or highly uncertain (*Supplementary file 1q-1s*). Overall, there was little evidence of mediation.

## Effect of covariates on outcomes

Of the included covariates, many were associated with outcomes. For adults (*Figure 4*; *Supplementary file 1a-1m*), age was positively associated with all negative health outcomes except respiratory illness as well as depression and social conflict. Male sex was associated with increased blood pressure but lower depression, conflicts, non-social problems, urinary cortisol, infection illness, and total morbidity, and with better self-rated health. Increasing distance from the market town was associated with increased blood pressure, more conflicts, respiratory illness, and gastrointestinal illness, as well as lower BMI. However, it was also associated with lower depression and urinary cortisol. Community size

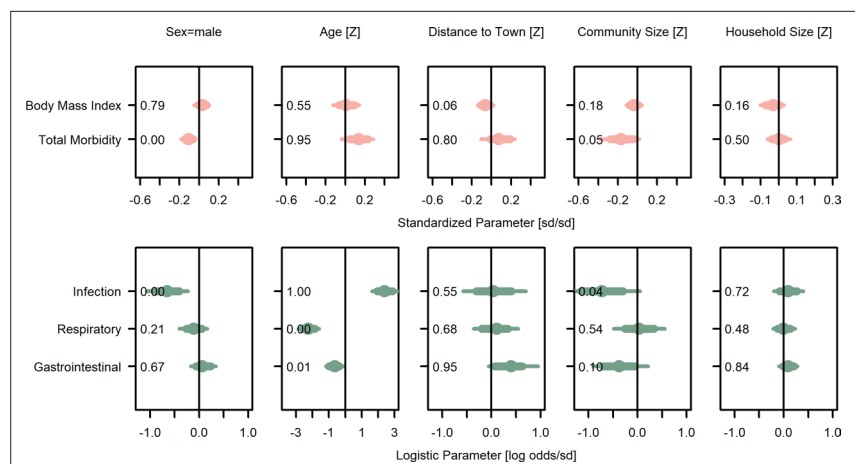

**Figure 5.** Covariate posterior parameter values for models with juveniles (≤15 years). Points are posterior medians and lines are 75% (thick) and 95% (thin) highest posterior density intervals. Numbers in each panel represent the proportion of the posterior distribution that is greater than zero ($P_{>0}$). Full models are given in **Supplementary file 1n and o**. Rough categories of dependent variables (continuous health outcomes and binary health outcomes) are distinguished by rows and colors. For the first row, the outcomes are measured as Z-scores, the bottom row as log odds.

was generally associated with more positive psychological and social variables, but also higher blood pressure and infection. Household size was associated with worse psychological and social condition, with the exception of labor partners, which were higher for large households. Results for juveniles largely reflect similar associations (**Figure 5**; **Supplementary file 1n and o**).

In some cases, the inclusion of covariates improved model $R^2$ statistics, though in many models changes in fit were negligible (**Supplementary file 1a-1o**). In general, the inclusion of covariates reduced the variance attributable to random effects for individual, household, and community. Posterior distributions for wealth and inequality associations were all similar whether covariates were included or excluded (i.e., the posteriors overlap substantially), though there were some minor differences between the posterior means that were largely inconsequential for inference.

## Is there evidence for more complex wealth-health associations?

Finally, we conducted several post-hoc tests to examine whether wealth-health associations were contingent on sex or whether relative wealth effects were contingent on levels of inequality and vice versa. For example, inequality could trigger increased stress and competitiveness only in men given a history of higher reproductive skew in males (**Daly, 2016**) and inequality might affect the wealthier and poorer differently (P2b), that is, poorer individuals may fare even worse in more unequal contexts. For this reason, we included wealth-by-inequality, wealth-by-sex, or inequality-by-sex interactions in models. A number of models favored interactions though there was little consistency across outcomes (**Figure 6**; **Supplementary file 1p**). For depression, systolic and diastolic blood pressure, and self-rated health, poorer men showed worse outcomes than wealthier men, though there was little effect of wealth for women. In contrast, poorer women reported more non-social problems. Poor individuals showed both increased conflicts and reduced labor partners in unequal places, while wealthier individuals reported more conflicts and fewer labor partners in equal communities. In unequal communities, wealth had little effect on respiratory illness, while in more equal places, wealthier individuals were less likely to be diagnosed with respiratory illness. Contra P2b, there was no consistent indication that inequality was worse for poorer individuals, while males were somewhat more affected by being poor.

## Discussion

We tested whether within-community relative wealth, community wealth, and community-level wealth inequality were associated with a broad range of psychological, social, and health outcomes in a large sample of households and communities in a relatively egalitarian small-scale subsistence society.

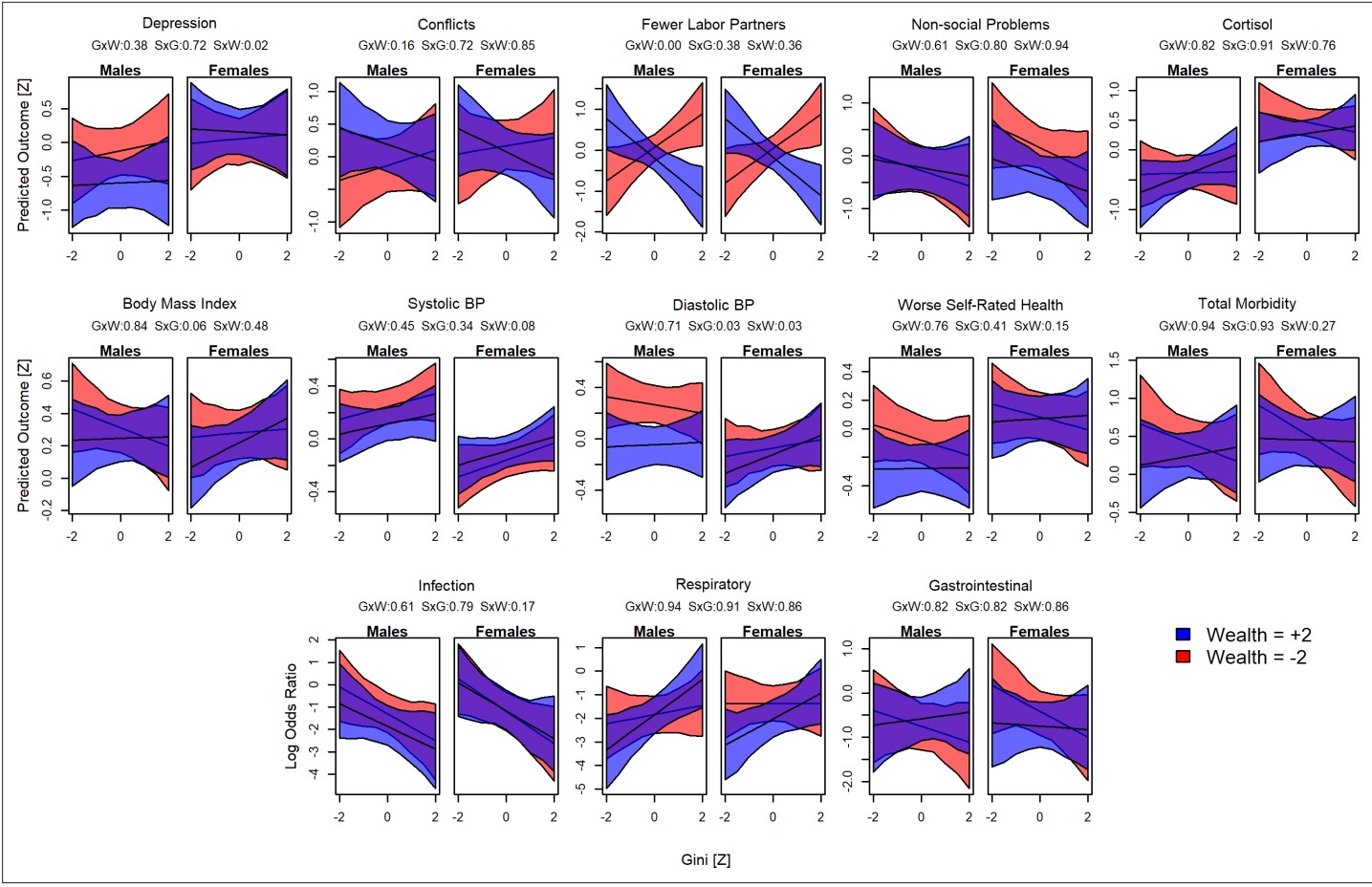

**Figure 6.** Interactions between sex, wealth, and inequality. Plots show the predicted values for each outcome and Gini Z-score. Red shading indicates poorer individuals (wealth Z = –2), blue indicates wealthier individuals (Z = 2). For each model, the proportion of the posterior >0 is shown in the numbers above: GxW: Gini × Wealth; SxG = Sex × Gini; SxW = Sex × Wealth.

Overall, our results showed substantial heterogeneity in terms of the direction and magnitude of associations between wealth, wealth inequality, and health, which contrasts with the more consistent socio-economic health gradients in high-income countries. Nevertheless, some findings supported an association between wealth or inequality and health outcomes, though these associations were not mediated by psychosocial factors.

Consistent with the prediction that higher relative position in a socio-economic hierarchy improves outcomes (P1), we found that household wealth relative to others in the community, capturing one's rank within the local socio-economic hierarchy, was associated with lower blood pressure, and for juveniles, lower total morbidity and fewer respiratory infections. Relative household wealth was also generally associated with better health and psychosocial outcomes, but with more uncertainty in the posterior estimates, and for juveniles, relative household wealth was associated with increased gastro-intestinal illness. Community mean wealth, capturing the absolute access to resources of households within that community, was also strongly associated with lower blood pressure for adults, but there was high uncertainty in estimates for other outcomes. Conversely, in support of P2a inequality was associated with higher blood pressure in adults and more respiratory disease in both adults and juveniles. It was also associated with lower BMI in juveniles, which in this energetically limited population likely represents a negative outcome. However, contra P2a inequality was also associated with lower levels of other infections (mostly fungal and yeast infections and lice) as well as fewer non-social problems, and there were several null results (*Figures 2 and 3*). Although most effect sizes were weak to moderate (most Cohen's *d* < 0.2), these statistically weak results could still have significant biological and clinical impacts, as elaborated below.

The finding that higher inequality associated with greater likelihood of respiratory disease is perhaps the most significant in terms of well-being and biological fitness. Respiratory illness is the leading cause of mortality at all ages in this population (*Gurven et al., 2007*) and continues to be a major source of morbidity. The likelihood of being diagnosed with respiratory illness was predicted to differ greater than threefold, 8–28%, between the least and most unequal communities indicating substantial fitness costs to inequality. However, this effect of inequality appears to primarily affect wealthy individuals, bringing their prevalence up to the level of poorer individuals (*Figure 6*). In one Tsimane community (with relatively high average income compared to other communities), *von Rueden et al., 2014* found lower risk of respiratory infection among influential men but no effect on respiratory infection (though trending in direction of higher risk) for men with higher income. With current data, we cannot determine the mechanism responsible for this association between inequality and respiratory disease. The association could reflect differences in immune function as suggested by other research on psychosocial influences on infectious disease (*Aiello et al., 2018*; *Chen and Miller, 2013*; *McDade et al., 2016*), despite a lack of evidence for psychosocial mediation here. The association between inequality and respiratory disease could also be spurious, despite our best efforts to control for relevant covariates, or it could reflect differences in exposure not captured by distance to town or community size (such as population density or frequency of contact with outsiders); in this context, it should be noted that effects of inequality on health are arguably only expected for outcomes for which there is a socio-economic gradient in the first place (*Pickett and Wilkinson, 2015*), which was not the case for respiratory disease here.

One of the strongest, most certain and most consistent associations of wealth (both household and community level) and inequality was with blood pressure, a major contributor to chronic disease in high-income countries. There was a clear socio-economic gradient in blood pressure within and between communities, and blood pressure was higher in more unequal communities. These effects were observed primarily in men. While most Tsimane are not hypertensive and do not have heart disease (*Gurven et al., 2012*; *Kaplan et al., 2017*), the predicted effects of wealth and inequality on blood pressure were substantial: systolic blood pressure was predicted to increase by 0.32 SD (i.e., 4.0 mmHg) and diastolic blood pressure by 0.40 SD (3.7 mmHg) in the most unequal compared to the most equal communities; conversely, wealth was protective such that the lowest blood pressures were predicted for people in the richest communities (7.4 mmHg systolic and 4.2 mmHg diastolic lower) and the richest households within communities (1.0 mmHg systolic and 3.1 mmHg diastolic lower). In high-income countries, such changes in blood pressure correspond to as much as a 10% change in the risk of major cardiovascular disease events (see Figure 2 in *Ettehad et al., 2016*). Among the Tsimane, it corresponds to as much as 40 years of age-related increases in blood pressure (*Gurven et al., 2012*). As novel, obesogenic foods enter the Tsimane diet (*Kraft et al., 2018*), market integration increases stress (*Konečná and Urlacher, 2017*; *von Rueden et al., 2014*), sanitation improves (*Dinkel et al., 2020*), and protective lifestyle factors like physical activity and helminth infections are changing (*Gurven et al., 2013*; *Gurven et al., 2016*), people in unequal communities, especially the poor (see *Figure 6*), may be at increasingly greater risk of chronic disease. Increases in blood pressure with modernization have also been reported in many other subsistence populations, and may partly stem from stress caused by integrating into a dominant culture (*Dressler, 1999*; *Konečná and Urlacher, 2017*). In this context, it is also worth noting that while the range of our village-level Gini values (0.15–0.43) was similar to that of *income* inequality among high-income countries (e.g., Denmark: 0.24; USA: 0.45), it was considerably lower than the range of *wealth* inequality in these countries (e.g., Japan: 0.55; USA: 0.81 [*Nowatzki, 2012*]). Thus, the reported associations between wealth/inequality and blood pressure may still be relatively harmless for the Tsimane, but lay the foundation for chronic disease under more mismatched conditions.

An alternative interpretation for some of these associations may be that causality is reversed, with poor health leading to less wealth or exacerbated inequality. On the face of it, this seems plausible for respiratory illness, which reduces work productivity. However, the fact that we see no direct association with wealth for adults, and only an association with inequality, seems to argue against such a mechanism. We did find an association between wealth and respiratory disease for juveniles – perhaps having sicker children puts some strain on wealth accumulation. For blood pressure, it is harder to imagine how reverse causality might occur since the blood pressure changes we observed are unlikely to affect wealth. Regardless, a limitation of our data is that we cannot determine the direction of

causation given our cross-sectional design. Other confounds might also be possible, for example, if people preferentially assort by health or wealth by moving between villages.

Beyond respiratory disease and blood pressure, many associations were inconclusive. This heterogeneous picture may seem surprising given robust directional findings from studies in high-income countries, especially for SES health gradients. One possibility for this difference is that hierarchy-stress associations produce more consistent health effects in an epidemiological context characterized by chronic, rather than infectious, disease. As argued above, our finding that one of the most consistent wealth-health associations was with blood pressure would support this argument since hypertension is a risk factor for most chronic diseases and consistently associated with socio-economic position and inequality in high-income countries (*Kim et al., 2008*; *Shahu et al., 2019*), but unlikely to be harmful for most Tsimane (*Gurven et al., 2012*; *Kaplan et al., 2017*). However, there are also consistent associations between socio-economic position and infectious disease in high-income countries (*Aiello et al., 2018*; *Snyder-Mackler et al., 2020*), suggesting that epidemiological context alone does not account for inconsistent results.

Another possible source of heterogeneity is the scale at which relative wealth and inequality are measured. Literature reviews suggest that at an international scale as many as 83% of studies find associations, while in studies of areas the size of neighborhoods, only 45% find associations (*Kondo et al., 2012*; *Pickett and Wilkinson, 2015*; *Wilkinson and Pickett, 2006*). *Pickett and Wilkinson, 2015* suggest that this heterogeneity reflects the scale at which inequality is perceived as most salient. Here, we assessed relative wealth and inequality at the scale of the residential community, a salient arena of daily cooperation and competition (*Alami et al., 2020*; *Gurven et al., 2015*; *Gurven et al., 2008b*; *Jaeggi et al., 2016*; *von Rueden et al., 2008*; *von Rueden et al., 2014*). This local level is also similar in scale to group-level hierarchies in other social species that show hierarchy-associated stress responses (*Sapolsky, 2005*; *Snyder-Mackler et al., 2016*; *Tung et al., 2012*). Furthermore, substituting community-relative wealth with wealth relative to the whole Tsimane population made little difference for results (*Supplementary file 1a-1m*), suggesting that the choice of scale within this relatively small-scale society did not matter. Modern technologies, such as television, may upset these comparisons and the functioning of hierarchy-related adaptations by making the global seem local; however, few Tsimane have regular access to television and other media. Nevertheless, it is possible that at least some Tsimane perceive inequality in reference to the local non-Tsimane population, or other regions of Bolivia, which was not captured by our study. Interacting with members of the dominant culture can be a source of stress (*Dressler, 1999*; *Konečná and Urlacher, 2017*; *von Rueden et al., 2014*), even if the Tsimane are arguably doing fairly well financially compared to other rural Bolivians (*Godoy et al., 2007*). Thus, we might not have been able to capture a relevant scale of comparison for some people, which could explain why associations at other scales were less consistent. However, this argument also applies to studies in high-income countries – where the relevant scales could be anything from neighborhoods to countries – and does not necessarily explain why results were inconsistent (as opposed to simply weaker) when measured at a less salient scale.

Finally, another explanation for heterogeneous associations is that our measure of household wealth may capture several distinct dimensions of socio-economic status, with partly orthogonal effects on health. On the one hand, greater wealth affords more respect and influence within communities, which is associated with lower cortisol and better health among the Tsimane (*von Rueden et al., 2014*) and elsewhere (*Decker, 2000*); this is likely the dimension captured by our subjective status data. On the other hand, household wealth is accumulated through participation in the market economy, which is associated with greater stress – higher cortisol, blood pressure – among the Tsimane (*von Rueden et al., 2014*) and elsewhere (*Dressler, 1999*; *Konečná and Urlacher, 2017*). The risks of different infectious diseases may also vary along these dimensions, with people who more frequently visit town and interact with outsiders possibly being more exposed to respiratory pathogens (*Kaplan et al., 2020*). Thus, household wealth may in part be inconsistently associated with health because of these opposing processes.

Several psychosocial variables were directly associated with health. Conflicts and depression were associated with lower BMI and blood pressure, perhaps indicating the effects of stress or lack of access to resources (depression is associated with low productivity among the Tsimane; *Stieglitz et al., 2014*). Depression and non-social problems were associated with worse self-rated health, again possibly via stress or direct effects of resource availability. However, associations between wealth or

inequality and health outcomes were not mediated when including psychosocial variables in models (P3), and there was almost no evidence for indirect effects proceeding through these pathways. An obvious limitation is that our sample sizes for the mediation analysis were smaller than for other analyses (*Supplementary file 1q-1s*), though most were still large enough to capture any meaningful effect. It is also possible that our measures of psychosocial stress were inadequate, for example, a single urinary cortisol measure likely captures overall differences in cortisol excretion (*Yehuda et al., 2003*), but does not capture changes in diurnal cortisol patterns that are typically associated with chronic stress (*García et al., 2017*; *Miller et al., 2007*). But the lack of mediation found here may also point to more nuanced mechanisms such as changes in physical activity related to different subsistence strategies or other lifestyle factors not accounted for here. For subsistence societies experiencing socio-economic change, whether relative status increases, decreases, or has no effect on stress and health may depend on the status measure and its association with social support. A study of four Tsimane communities found that influential men with greater social support had lower cortisol (*von Rueden et al., 2014*), but higher cash income associated with higher cortisol (*von Rueden et al., 2014*; see also *Konečná and Urlacher, 2017*). In another study of the Tsimane, higher incomes predicted lower BMIs, unless individuals had relatively more social support (*Brabec et al., 2007*). It therefore remains unclear what mechanisms were responsible for the wealth-health associations found here, though hierarchy is known to affect immune function, and thereby infectious disease morbidity independently of stress and associated HPA activity (*Aiello et al., 2018*; *Miller et al., 2011*; *Snyder-Mackler et al., 2020*; *Snyder-Mackler et al., 2016*).

In sum, we present the most comprehensive test of hierarchy-health associations in a subsistence society to date. In support of an evolutionary argument that conceptualizes hierarchy-health effects as stemming from evolved reaction norms adjusting people's behavior and physiology to the rank and local competitive regime they find themselves in (*Daly and Wilson, 1997*; *Griskevicius et al., 2011*; *Pepper and Nettle, 2014*), we found that wealth and inequality were associated with several health outcomes, though other associations were negligible or in the opposite direction to that predicted. In support of the argument that most hierarchy-health effects in high-income countries are caused by evolutionary mismatch (*Sapolsky, 2004*), we found that inequality was associated with blood pressure but in a range unlikely to affect health; however, this association could lead to hypertension, cardiovascular and metabolic disease as inequality further increases due to increased market integration and/or as novel foods and lifestyle factors enter the population (*Gurven et al., 2012*; *Gurven et al., 2016*; *Kaplan et al., 2017*; *Kraft et al., 2018*). Our study thus contributes to an evolutionary approach to public health that considers tradeoffs and mismatch as important links between socio-ecology, lifestyle, and health (*Eaton et al., 2002*; *Wells et al., 2017*).

## Materials and methods

### Data collection and preparation

All data were collected under the auspices of the Tsimane Health and Life History Project (THLHP) (*Gurven et al., 2017*) by a team of Bolivian medical professionals and Tsimane researchers.

### Wealth and wealth inequality

Wealth data were collected in 2006–2007 and 2013. Here, we only included wealth data collected prior to a rare catastrophic flood in February 2014 that destroyed crops and household goods in the vast majority of Tsimane communities (*Trumble et al., 2018*). *Figure 1—figure supplement 1* summarizes how many individuals were included in the sample, out of all individuals ever sampled by the THLHP. Household wealth was assessed through an inventory of commonly owned items including traditional goods, that is, items manufactured from local organic materials (e.g., canoes, bows and arrows), market goods, that is, industrially produced items obtained through trade or purchase (e.g., bicycles, motorbikes), and livestock (e.g., pigs, cows), which were subsequently converted into their local market value in Bolivianos and summed (*Figure 1*).

Objective household wealth arguably provides only an indirect measure of people's subjective wealth and status (*Norton, 2013*), but these data were most widely available for this study. Furthermore, household wealth correlated significantly, albeit weakly, with subjective status (*Amir et al., 2019*; *Woolard et al., 2019*; $r = 0.17$, df = 147, p<0.05) and subjective wealth rank ($r = 0.29$, df =

150, p<0.001). Previous work among the Tsimane (*Undurraga et al., 2016*) has also shown that more visible forms of wealth, such as the household items counted here, influenced subjective health more than less visible forms of wealth, such as the size of cultivated fields. To prevent differences in age sampling between villages from affecting wealth and inequality estimates, we followed *Borgerhoff Mulder et al., 2009* and adjusted wealth values for the age of the head of household by fitting generalized additive models for location scale and shape (GAMLSS) to the distribution of wealth-by-age to obtain wealth-by-age Z-scores. Wealth Z-scores derived from GAMLSS, representing centile values, were used in all analyses in part because wealth was skewed in distribution, and also expected to have diminishing returns at higher values (i.e., 100 Bolivianos are worth more to a poor individual than a wealthy one). However, to determine whether Z-scoring with GAMLSS altered results by normalizing the shape of the wealth distribution, we also repeated analyses with standardized wealth (i.e., [household wealth – population-average wealth]/standard deviation of population-average wealth), which preserves the skew. There were no qualitative differences in inference between the two methods, largely because Z-scoring with GAMLSS primarily affects outliers on the far high end of the distribution. Note that 'Z-score' can have two slightly different meanings; for wealth and BMI (see below), we generally mean centile values from GAMLSS unless otherwise noted, for all other variables Z-scores refer simply to standardized values (i.e., $[x - mean(x)]/sd(x)$).

Mean wealth and wealth inequality at the community level (for communities with ≥ 9 households) were calculated after converting wealth Z-scores back into equivalent values in Bolivianos at age 50 (see *Figure 1*). We used the Gini coefficient to measure inequality; other inequality measures (e.g., median share, 90/10 ratio) generally correlate highly ($r > 0.94$) with Gini (*Kondo et al., 2009*) and were therefore not considered. In other studies, local scales of measuring inequality, such as at the community level used here, tend to produce smaller effects on health than those at larger scales, such as states or countries (*Kondo et al., 2009*; *Wilkinson and Pickett, 2006*). In the Tsimane context, it is unclear whether that will be the case given low residential mobility and concentration of work and socializing within communities. However, Tsimane visit other communities and sporadically engage in market-based interaction with non-Tsimane, and comparisons with wealthier neighbors can contribute to Tsimane status aspirations (*Schultz, 2019*). Nevertheless, as mentioned above (Study population), we consider the community to be the most relevant arena for status competition among Tsimane (though substituting community-relative wealth with population-relative wealth made little difference; see *Supplementary file 1a-1m*). Note that most studies on health effects of inequality use *income* inequality (but see *Nowatzki, 2012*), which is less unequally distributed than wealth. Cash income among the Tsimane during this study period was sporadic and many households may have no income in a given sampling period, which leads to overestimated Ginis. We therefore preferred wealth and wealth inequality as a more reliable measure of households' long-term access to resources and its distribution.

## Psychological, social, and health variables

The THLHP has been recording biomedical and anthropological data during roughly annual medical examinations and interviews by THLHP physicians and research assistants on an increasing number of communities since 2002. Here, we included any data collected within 2 years of an individual's wealth data, that is, the potential range of data was 2004–2009 and 2011–2015. *Table 1* summarizes how many individuals out of all the ones with wealth data (see also *Figure 1—figure supplement 1*) were included for each outcome variable.

Depressive symptoms were measured using an adapted 18-item questionnaire (*Stieglitz et al., 2014*), the responses to which were summed to yield an overall depression score. The same interview also asked whether participants experienced conflicts with several kinds of social partners as well as non-social problems, such as food insecurity, illness, or debt; affirmative answers were summed to yield a composite measure of social conflicts and non-social problems, respectively. A household's cooperation network was measured as the number of people from different households who helped in that household's fields in a given year. Cortisol was measured in first-morning urine using enzyme-linked immunosorbent assays and corrected for specific gravity (see *von Rueden et al., 2014* for details). BMI Z-scores were calculated by GAMLSS using Tsimane-specific growth curves (*Blackwell et al., 2016b*) (R package at: https://github.com/adblackwell/localgrowth (*Blackwell, 2021*) copy archived at swh:1:rev:81ce799bdc6da90d48e5ad8afd6ad0f3b19494d2) as well as the total distribution of

Tsimane adult BMIs, representing deviations from the local population average for a given age and sex. Diastolic and systolic blood pressures were measured by THLHP physicians using an aneroid sphygmomanometer. Self-rated general health was measured using a five-point scale from ('very bad' [1] to 'excellent' [5]). Morbidity at the time of the medical check-up was assessed by physicians using the International Classification of Disease, 10th edition (ICD-10 classification) and then grouped into 18 clinically meaningful categories following the Clinical Classifications System (CCS) (https://www.hcup-us.ahrq.gov/toolssoftware/ccs/ccsfactsheet.jsp); morbidities in any of these categories were summed to give a total morbidity score potentially ranging from 0 (no morbidities) to 18 (at least one morbidity in each category). In addition, we also examined the presence/absence of infectious and parasitic diseases (CCS 1, hereafter 'infections'), diseases of the respiratory system (CCS 8, 'respiratory illness'), and diseases of the digestive system (CCS 9, 'gastrointestinal illness'), which represent the most common causes of morbidity and mortality in this population (*Gurven et al., 2020*; *Gurven et al., 2007*). See *Table 2* for examples of the six most common diagnoses in these three categories. Distance to the town of San Borja was measured as nearest route (whether by river or road) from the center of the community and provides a proxy for access to modern amenities. Community size and household size were summarized from complete population censuses conducted regularly by the THLHP. Thus, they include all individuals, not just those sampled for wealth or other covariates.

## Data analysis

Prior to analysis, all variables except binary variables were standardized into Z-scores. Urinary cortisol was log transformed prior to standardization to reduce skew, as is common practice. All outcomes were modeled as Gaussian, except the presence/absence of specific morbidities (Bernoulli). Each analysis modeled an individual-level outcome as a function of individual-, household-, and community-level characteristics (*Table 1*). Thus, we fit the following base model for each outcome:

Outcome$_{ijkl}$ ~ β$_0$ + (β$_1$ * Sex$_j$) + (β$_2$ * Age$_j$)+ (β$_3$ * relative household wealth$_k$) + (β$_4$ * Community-level Gini$_l$) + (β$_5$ * Community-level mean wealth$_l$) + (β$_6$ * Community Size$_l$) + (β$_7$ * Distance of community to market town$_l$) + (β$_8$ * Household Size$_l$) $u_j$ + $u_k$ + $_{ul}$ + $e_{ijkl}$

wherein the subscripts denote measurement $i$, individual $j$, household $k$, and community $l$, respectively. β$_0$ is the intercept, all other βs are slopes, $u$s are random intercepts, and $e$ is the residual error (not available for Bernoulli responses). Variance inflation factors (VIFs) indicated virtually no collinearity among predictors (all VIFs < 3).

In order to test whether potential wealth-health associations were mediated by psychosocial stress, we reran all health models (blood pressure, self-rated health, total morbidity, infections, respiratory and gastrointestinal illness) with pertinent psychosocial variables as covariates and used the *mediation* function in the *sjstats* package (*Lüdecke, 2021*) to estimate direct and indirect effect. In addition, we also ran a series of exploratory analyses in which we added interaction terms.

We used Bayesian multilevel models fit with the *brms* package v. 2.13.5 (*Bürkner, 2017*) in R 4.0.2 for all analyses. All models used regularizing priors (fixed effects: normal, mean = 0, SD = 1; random effects: half-Cauchy, location = 0, scale = 2), which imposes conservatism on parameter estimates and reduces the risk of inferential errors (*Gelman et al., 2013*; *McElreath, 2020*). All models converged well as assessed by inspecting trace plots and standard diagnostics (all Rhat < 1.01). All data and R code are available at https://doi.org/10.5281/zenodo.4567498 with any updates at https://github.com/adblackwell/wealthinequality (*Jaeggi et al., 2021*, copy archived at swh:1:rev:da16ac6b20732fe1939478450d81ac32fdcce202).

## Acknowledgements

We thank the Tsimane for their generous participation and years of collaboration, and THLHP personnel for their herculean efforts and dedication in data collection. We thank Erik Ringen, Jordan Martin, participants in the Club EvMed series, as well as Milagros Ruiz, Prabhat Jha, and one anonymous reviewer for comments and suggestions that greatly improved this article.

The Tsimane Health and Life History Project has been funded by NSF (BCS0136274, BCS0422690) and NIH/NIA (RF1AG054442, R01AG024119, R56AG024119). AJ was supported by the Swiss NSF (PBZHP3-133443) and by the SAGE Center for the Study of the Mind. JS acknowledges Institute for Advanced Study Toulouse funding from the French National Research Agency (ANR) under grant ANR-17-EURE-0010 (*Investissements d'Avenir* program). BAB acknowledges research funding from

the Max Planck Society, through the Department of Human Behavior, Ecology and Culture. The authors report no conflicts of interests.

## Additional information

### Funding

| Funder | Grant reference number | Author |
|---|---|---|
| Schweizerischer Nationalfonds zur Förderung der Wissenschaftlichen Forschung | PBZHP3-133443 | Adrian V Jaeggi |
| National Science Foundation | BCS0136274 | Hillard Kaplan |
| National Science Foundation | BCS0422690 | Michael Gurven |
| National Institutes of Health | R01AG024119 | Hillard Kaplan Michael Gurven |
| National Institutes of Health | RF1AG054442 | Hillard Kaplan Michael Gurven |
| National Institutes of Health | R56AG024119 | Hillard Kaplan Michael Gurven |

The funders had no role in study design, data collection and interpretation, or the decision to submit the work for publication.

### Author contributions

Adrian V Jaeggi, Conceptualization, Data curation, Formal analysis, Methodology, Visualization, Writing – original draft, Writing – review and editing; Aaron D Blackwell, Conceptualization, Data curation, Formal analysis, Funding acquisition, Methodology, Project administration, Resources, Visualization, Writing – original draft, Writing – review and editing; Christopher von Rueden, Benjamin C Trumble, Jonathan Stieglitz, Hillard Kaplan, Data curation, Funding acquisition, Project administration, Writing – original draft; Angela R Garcia, Thomas S Kraft, Data curation, Writing – original draft; Bret A Beheim, Paul L Hooper, Data curation, Writing – review and editing; Michael Gurven, Conceptualization, Data curation, Funding acquisition, Project administration, Resources, Writing – original draft, Writing – review and editing

### Author ORCIDs

Adrian V Jaeggi (ID) http://orcid.org/0000-0003-1695-0388
Aaron D Blackwell (ID) http://orcid.org/0000-0002-5871-9865
Jonathan Stieglitz (ID) http://orcid.org/0000-0001-5985-9643
Angela R Garcia (ID) http://orcid.org/0000-0002-6685-5533
Michael Gurven (ID) http://orcid.org/0000-0002-5661-527X

### Ethics

Human subjects: Institutional Review Board approval was granted by UNM (HRRC # 07-157) and UCSB (# 3-16- 0766), as was informed consent at three levels: (1) Tsimane government that oversees research projects, (2) village leaders and community meetings, and (3) study participants.

### Decision letter and Author response

Decision letter https://doi.org/10.7554/eLife.59437.sa1
Author response https://doi.org/10.7554/eLife.59437.sa2

## Additional files

### Supplementary files

• Supplementary file 1. This file contains the following tables with additional information on the statistical models. a: Model summary – depression. b: Model summary – social conflicts. c: Model summary – fewer labor partners. d: Model summary – non-social problems. e: Model summary – cortisol. f: Model summary – BMI. g: Model summary – systolic blood pressure. h: Model summary – diastolic blood pressure. i: Model summary – worse self-rated health. j: Model summary – total morbidity. k: Model summary – infections. l: Model summary – respiratory illness. m: Model summary – gastrointestinal illness. n: Gaussian model summaries for juveniles. o: Logistic model summaries for juveniles. p: Overview of exploratory interaction effects. q: Mediation of wealth effects. r: Mediation of inequality effects. s: Mediation of mean community wealth effects.

• Transparent reporting form

### Data availability

All data and R code are available at https://doi.org/10.5281/zenodo.4567498 with any updates at https://github.com/adblackwell/wealthinequality (copy archived at https://archive.softwareheritage.org/swh:1:rev:da16ac6b20732fe1939478450d81ac32fdcce202).

The following dataset was generated:

| Author(s) | Year | Dataset title | Dataset URL | Database and Identifier |
|---|---|---|---|---|
| Jaeggi AV, Blackwell AD, von Rueden C, Trumble BC, Stieglitz J, Garcia AR, Kraft TS, Beheim BA, Hooper PL, Kaplan H, Gurven M | 2021 | | https://zenodo.org/record/4567498#.YMNxvZNKi3l | Zenodo, 10.5281/zenodo.4567498 |

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

## Appendix 1

### Causal relationships assumed by mediation analysis

The causal relationships between independent variable, mediator, and dependent variable assumed by standard mediation analysis are depicted in *Appendix 1—figure 1*.

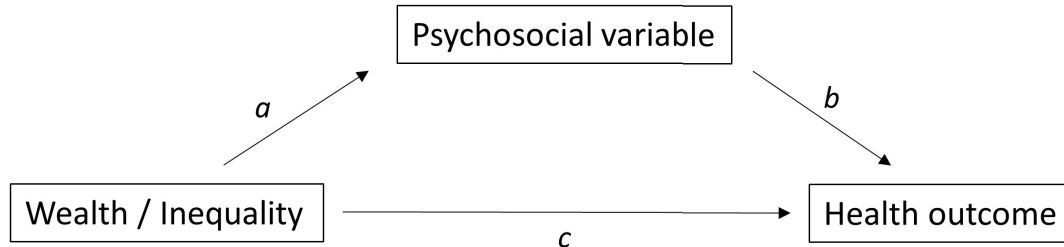

**Appendix 1—figure 1.** Causal relationships assumed by the mediation analyses.

In our case, the independent variables are absolute wealth, relative wealth, and inequality, the mediators are the psychosocial variables, and the dependent variables are the health outcomes (see *Supplementary file 1q-1s*).

The next causal diagram, *Appendix 1—figure 2* highlights a potential problem of this approach, which treats each mediator separately even though several mediators are present.

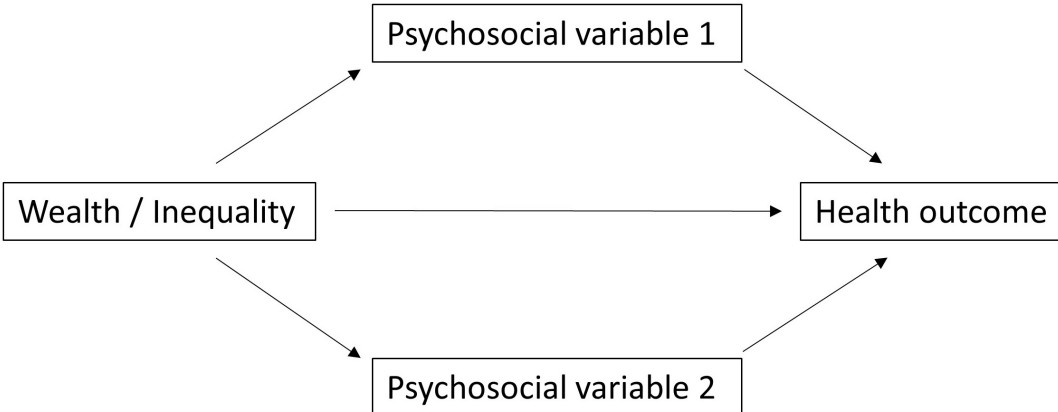

**Appendix 1—figure 2.** Causal diagram highlighting multiple mediation.

For example, it could be that wealth/inequality cause both higher levels of depression and higher levels of cortisol. In this scenario, leaving one mediator out prevents us from accurately estimating the 'direct effect' of our independent variable on health outcomes, limiting us only to the total effect absent the mediating pathway.

Furthermore, our results could also be influenced by collider bias in the case that different mediating variables are themselves causally linked, as in *Appendix 1—figure 3*.

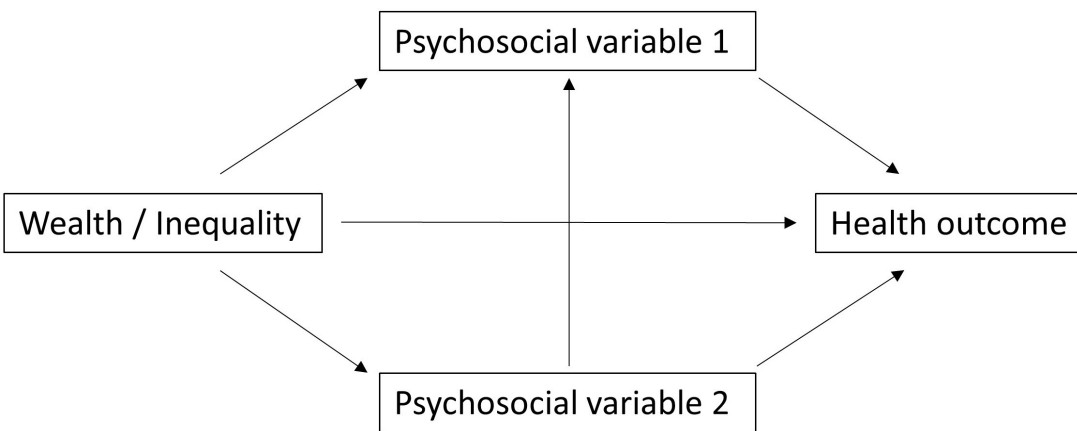

**Appendix 1—figure 3.** Causal diagram highlighting multiple mediation and collider bias.

Following the example above, it could be that cortisol also independently causes higher levels of depression (or vice versa). In this situation, the mediating psychosocial variable 1 functions as a collider between wealth/inequality and psychosocial variable 2, and we cannot properly assess the impact of our independent variables on the outcome without conditioning on all mediation variables simultaneously.

Despite these potential problems, we preferred the present approach of treating each mediator independently because including all mediators in the same analysis would have required imputing most of the values, because the sample overlap was small. Furthermore, we have good reasons to believe that this approach would not change inference. Namely, in a previous version of the mediation analysis (https://www.medrxiv.org/content/10.1101/2020.06.11.20121889v1) we performed a principle components analysis on all psychosocial variables and found that they were relatively uncorrelated, as the main principle component only contained depression loadings. Furthermore, we had included depression, non-social problems, and cortisol (with imputation) in the same models with the same inference as in the present version of the analysis: there was no convincing evidence for mediation.

