## [Decision Letter]

**Acceptance summary:**

This novel study tests the associations between wealth and wealth inequality with a range of health outcomes in the Tsimane. While these associations have been widely evaluated in urbanised populations around the world, the present study provides unique evidence by studying a small-scale subsistence society in Bolivia.

**Decision letter after peer review:**

Thank you for submitting your article "Relative wealth and inequality associate with health in a small-scale subsistence society" for consideration by *eLife*. Your article has been reviewed by 3 peer reviewers, one of whom is a member of our Board of Reviewing Editors, and the evaluation has been overseen by George Perry as the Senior Editor. The reviewers have opted to remain anonymous.

The reviewers have discussed the reviews with one another and the Reviewing Editor has drafted this decision to help you prepare a revised submission.

As the editors have judged that your manuscript is of interest, but as described below that additional analyses are required before it is published, we would like to draw your attention to changes in our revision policy that we have made in response to COVID-19 (https://elifesciences.org/articles/57162). First, because many researchers have temporarily lost access to the labs, we will give authors as much time as they need to submit revised manuscripts. We are also offering, if you choose, to post the manuscript to bioRxiv (if it is not already there) along with this decision letter and a formal designation that the manuscript is "in revision at *eLife*". Please let us know if you would like to pursue this option. (If your work is more suitable for medRxiv, you will need to post the preprint yourself, as the mechanisms for us to do so are still in development.)

Essential revisions:

1. The presentation of results are confusing to the reader. Since count scores, binary and continuous outcomes are examined, it would be easier if standardised β coefficients were reported and compared throughout the paper instead of switching between raw and exponentiated coefficients. The results currently use z-scores for continuous predictor variables to facilitate interpretation of some model coefficients. However, it's clear (Figure 1B) that z-scoring wealth distorts absolute wealth differentials: 1 unit differences in the z score correspond to very large differences in wealth for the wealthy, and very small differences in wealth for the poor. This could influence the modeling results (in a direction of diminishing true wealth effects) and affect comparisons about the influence of relative and absolute wealth (e.g., lines 275-276). Comparing model coefficients of standardizing betas instead of z-scores of the predictor variables may therefore be more accurate, especially for very skewed distributions. In addition, can some outcomes be reverse coded so that higher coefficients consistently reflect worse health? This change would forgo the need for the last column in Table 1.

2. There is insufficient discussion on the role of covariates in the hypothesised associations since only fully adjusted models are reported. It would be helpful to add some metrics, such as the reduction in chi square or likelihood ratio statistics with the main variable and then with all covariates. This would also inform whether there is residual confounding.

3. Since the study reports 26 associations (wealth and wealth inequality with 13 outcomes separately), it is difficult to glean overall conclusions from the large number of mixed findings. It may be helpful to present associations between wealth and wealth inequality with health, in turn, followed by how these hypothesised associations were attenuated by potential psychosocial factors. Since psychosocial variables are theorised as mediators, but are reported concurrently with the health outcomes, the causal inferences being made in the paper are unclear. Given the large number of results, it would be helpful to add subheadings to parse out different parts of the Results section.

4. This may vary by discipline, but it is inappropriate to test mediators by adjusting for them as covariates in epidemiology. Mediation is normally assessed using causal modelling techniques. Furthermore, the study appears to use a cross-sectional data set making it difficult to distinguish between cause and effect irrespective of methods used. Therefore, the study cannot provide any definitive mediational results. The Introduction nicely summarises the theoretical basis of a psychosocial stress pathway, so perhaps the authors could explore the potential role of psychosocial factors in wealth-health associations without making strong claims. By reporting that there was weak evidence of associations between wealth/wealth inequality with psychosocial factors (paths A) and between psychosocial factors and health outcomes (paths B), in turn, according to the Baron and Kenny approach, one could say that there was weak, albeit indirect, evidence of these factors as potential mediators. Given the hypothesised relations, it is clearer not to refer to psychosocial factors when referring to health outcomes and vice versa.

5. The results for community-level inequality measures were difficult to understand. Based on Figure 1, it looks like higher inequality communities have small samples, with the exception of Mission. Do these sample size differences affect the reliability of the inequality effect estimates? Related: are community-level measures based on the households sampled, or on the entire community including individuals who may not be in some of the analyses? Or are household samples exhaustive for these communities? In the inequality analysis, do measures based only on the households sampled (if non-exhaustive) affect whether the Gini coefficients are representative of the community (especially in small samples, e.g. n=9)?

6. We found it challenging to interpret some of the results without more detail on the modelling strategy. Even after reading the methods, it was unclear who was included in the analysis. For example, in Table 1 the age range is 0-89.7 years. Does this mean that infants, children, and adolescents are grouped in the analysis with adults? If so, what's the justification for this approach, given that these age groups may experience wealth, morbidity, and mortality risk in qualitatively different manners (not captured by modeling age as a covariate)? Page 8 claims that the study includes 1/3 of the Tsimane population so does the analytic sample include approx. 5,300 persons of the 16K?

7. For the morbidity data, were morbidities counted based on occurrence at the time of the medical check-up, or also retrospectively across some span of time? If the former, to what degree do you think this would introduce measurement error?

8. Since Bayesian imputation might lead to some errors in estimating effects, the authors should comment on this. Furthermore, statistical methods to deal with missing data are useful to a degree, but Methods should report sample sizes and the proportion and reason for such high levels of missing data. If we read Table 1 correctly, the number of adults with health data vary from 850 adults for self-rated health to 9000 for BMI? Data on psychosocial factors are also quite low – if missing data are so high, how reliable are these data? Is this why there was weak evidence on the role of psychosocial factors? The study should provide information on the data quality in Methods and reflect on these limitations in Discussion.

9. There was some discussion between reviewers as they had some differences in interpretation of the results. One reviewer found that wealth inequality-health effects were more substantial than wealth-health, which is contrary to the authors' interpretation. For example, if comparing Figures 2 and 3, wealth associations with health were mixed across outcomes (3 strong associations out of 8), but wealth inequality associations were more consistent (5 strong associations out of 8). If this interpretation is correct, it would be important to consider the relative inconsistency/consistency of associations between outcomes since the paper sometimes refer to key findings for "health" as a global metric. Among the paper's strengths is its inclusion of a wide range of unique health outcomes, making it important to highlight any differences in results between health outcomes and to discuss potential reasons why household wealth is a weaker predictor of systolic and diastolic blood pressure than wealth inequality, for example.

10. The limitation of the difficulty of disentangling association vs. causation and the potential of reverse causation (due to the cross-sectional design) needs to be addressed more directly and clearly. If inferential analyses cannot be made to support the causation case, the quality of the evidence being presented in the paper should be discussed more critically.

11. The paper is written in an evolutionary medicine framework, with contrasts drawn between a "trade-offs" model and a mismatch model. This is an interesting presentation, but I'm concerned that the mismatch model is essentially unfalsifiable here. If social gradients did not exist in the Tsimane, mismatch would probably be invoked. But based on the results presented here, they do, including for markers of cardiovascular health-and this is also presented as support for evolutionary mismatch. An equally likely interpretation is that responses to relative wealth and social inequality across different types of human societies are shared, counter to the mismatch hypothesis. Because mismatch can be supported whether or not health-wealth associations are found, P4a and P4b become somewhat poorly defined predictions. For P4b, it's also worth noting that social gradients in health are observed in high-income countries for both chronic and infectious disease (the current pandemic is a good example of the latter). The results are valuable to understanding the evolutionary history of social gradients in humans, but a more critical take on which hypotheses it can robustly test would improve the manuscript.

[Editors' note: further revisions were suggested prior to acceptance, as described below.]

Thank you for submitting your article "Do wealth and inequality associate with health in a small-scale subsistence society?" for consideration by *eLife*. Your article has been reviewed by 3 peer reviewers, including Milagros Ruiz as the Reviewing Editor and Reviewer #1, and the evaluation has been overseen by George Perry as the Senior Editor. The following individual involved in review of your submission has agreed to reveal their identity: Prabhat Jha (Reviewer #2).

The authors have addressed all points raised during the review. We commend the authors for their substantial efforts on the manuscript, including the

additional analyses. We have only one further essential revision, which we hope will be a straightforward addition to the manuscript:

1. In order to help readers understand the range of results (e.g., many observed associations were weak or uncertain), we recommend that the authors expand the discussion with a few statements about the high degree of heterogeneity in their findings explicitly, especially given the wealth of evidence from high-income developed nations which generally find strong and directionally consistent effects of SES on health.

*Reviewer #1:*

The authors have addressed all major theoretical and methodological points raised during the review. We appreciate the work of the authors to address all of the reviewer comments, including the additional analyses.

*Reviewer #2:*

I commend the authors for addressing all the review comments seriously. I have no further review comments.

*Reviewer #3:*

Overall, this is a strong revision that is highly responsive to previous reviewer comments and concerns. As before, I think the manuscript presents a timely, important analysis of the social determinants of health in small-scale societies, using a uniquely rich data set. Several substantive changes help improve clarity and readability from the original submission. In particular, I appreciate the decision to uniformly code the direction of health outcomes, straightforward statements about the authors' own interpretation of the data, and the inclusion of sufficient methodological detail for readers to also draw their own conclusions. The revision also does a much-improved job discussing the relationship of the analysis to evolutionary mismatch. I'm strongly supportive of publication.

At the same time, the results are still quite complex to work through, and many of the observed associations are weak or uncertain enough that it's unclear whether they're "real" (this is exacerbated somewhat by the many statistical tests, which raises questions about multiple hypothesis testing). To be clear, I do not think further or altered analyses are needed: the complexity in the results seems likely to reflect, at least in large part, real-world complexity in wealth-health relationships in this society. However, the authors might want to consider discussing the high degree of heterogeneity in their findings explicitly, especially considering the sense one gets from the literature on wealthy, developed nations, that SES effects on morbidity/mortality are often both strong and directionally consistent.

---

## [Author Response]

Essential revisions:1. The presentation of results are confusing to the reader. Since count scores, binary and continuous outcomes are examined, it would be easier if standardised β coefficients were reported and compared throughout the paper instead of switching between raw and exponentiated coefficients.

We agree that a better way to compare associations with binary and Gaussian outcome variables was desirable. We have now added Cohen’s d as a standardized measure of effect size for all coefficients, which makes effect sizes easily comparable. For binary outcomes we now also report the β values (on the log-odds scale) as well as the exponentiated values (odds ratios). Furthermore, in order to keep the text of the Results section consistent we have now defined criteria for referring to the strength of effect sizes and the certainty of associations (lines 350-356):

“For simplicity, we refer to effect sizes of d>0.2 as “strong”, those >0.1 as “moderate”, and consider the rest to be “weak” though potentially still suggestive of a general pattern. […] However, we encourage readers to use the full information on the posteriors to inform their own inference.”

The results currently use z-scores for continuous predictor variables to facilitate interpretation of some model coefficients. However, it's clear (Figure 1B) that z-scoring wealth distorts absolute wealth differentials: 1 unit differences in the z score correspond to very large differences in wealth for the wealthy, and very small differences in wealth for the poor. This could influence the modeling results (in a direction of diminishing true wealth effects) and affect comparisons about the influence of relative and absolute wealth (e.g., lines 275-276). Comparing model coefficients of standardizing betas instead of z-scores of the predictor variables may therefore be more accurate, especially for very skewed distributions.

The reviewers are correct that the distribution of absolute wealth is skewed, and that the use of wealth z-scores as calculated with GAMLSS, representing centiles, normalizes this skew (see left and right panels in Author response image 1). This is precisely the reason to use z-scores instead of absolute wealth. The logic is that we expect there to be diminishing returns per dollar in terms of absolute wealth, as wealth increases. For a poor person, a small increase in wealth would be expected to have a large effect, whereas for a wealthy person a similar increase in money would have virtually no effect. By using z-scores for relative wealth we normalize this effect of diminishing returns. Note that the Gini coefficients however were calculated based on the actual wealth values, not the z-scores, to reflect the actual inequality.

**Author response image 1. sa2fig1:** 

Nevertheless, we agree that the reviewers made a great point: it is possible that normalizing the wealth distribution misses some key information about wealth differentials that people might be sensitive to and that could therefore affect their behavior and health. We therefore ran an additional set of models with standardized wealth (i.e. household wealth – average wealth / standard deviation of wealth), which preserves the skewed distribution (see Figure above, middle panel). There were no meaningful qualitative changes in the results. We’ve added the following to the text to explain this approach (lines 753-760):“Wealth z-scores derived from GAMLSS, representing centile values, were used in all analyses in part because wealth was skewed in distribution, and also expected to have diminishing returns at higher values (i.e. 100 Bolivianos are worth more to a poor individual than a wealthy one). […] There were no qualitative differences in inference between the two methods, largely because z-scoring with GAMLSS primarily affects outliers on the far high end of the distribution.”

We also disambiguate the slightly different meanings of z scores (lines 760-763):

“Note that “zscore” can have two slightly different meanings; for wealth and BMI (see below), we generally mean centile values from GAMLSS unless otherwise noted, for all other variables Z scores refer simply to standardized values (i.e. [x – mean(x)] / sd(x))”.

In addition, can some outcomes be reverse coded so that higher coefficients consistently reflect worse health? This change would forgo the need for the last column in Table 1.

Thank you for this good suggestion. We’ve reverse coded the labor partners and self-rated health variables. This makes the tables and figures clearer as all positive associations now refer to beneficial outcomes and all negative ones to detrimental outcomes (though it does make the language in the text slightly more complicated).

2. There is insufficient discussion on the role of covariates in the hypothesised associations since only fully adjusted models are reported. It would be helpful to add some metrics, such as the reduction in chi square or likelihood ratio statistics with the main variable and then with all covariates. This would also inform whether there is residual confounding.

Thanks again for a good suggestion. We’ve added models with no covariates to the supplementary tables, i.e. only including household wealth and inequality, as well as a section and figures to the results discussing the effects of covariates (Section “*Effect of covariates on outcomes”*, Figures 4 and 5). Many covariates were associated with outcomes, but wealth and inequality (plus random effects) alone explained most of the variance in all models. We’ve also highlighted the effect of mean community wealth more in Figures 2 and 3 as well as in the results text (sections “*Is wealth related to …?”*).

3. Since the study reports 26 associations (wealth and wealth inequality with 13 outcomes separately), it is difficult to glean overall conclusions from the large number of mixed findings. It may be helpful to present associations between wealth and wealth inequality with health, in turn, followed by how these hypothesised associations were attenuated by potential psychosocial factors. Since psychosocial variables are theorised as mediators, but are reported concurrently with the health outcomes, the causal inferences being made in the paper are unclear. Given the large number of results, it would be helpful to add subheadings to parse out different parts of the Results section.

Thanks for this comment. We are keenly aware of the difficulty of presenting such a large number of analyses, and appreciate the suggestion of using subheadings and a more systematic reporting of results. We’ve now reorganized the Results section with several subheadings to break out the effects of wealth and inequality on physical and psychological and social variables, focusing on the health outcomes first. Specifically, the subheadings are:

– Is wealth related to health outcomes? (this includes household wealth and community wealth)– Is inequality related to health outcomes?– Is wealth related to psychosocial outcomes? (this includes household wealth and community wealth)– Is inequality related to psychosocial outcomes?– Do psychosocial variables mediate relationships between wealth or wealth inequality and health?– Effects of covariates on outcomes

We also present all associations within each subsection more systematically, e.g. “household wealth was associated … with reductions in all negative health outcomes except gastrointestinal illness”, prior to giving more detail on how strong and well supported the associations were. We also changed the main results figures to coefficient plots, which should help provide a better overview – the previously presented counterfactual plots have now been added as supplements to these coefficient plots. As a result of all these changes, and the addition of standardized effect sizes (Cohen’s d) and reverse coding (see above), the reader can now more easily and systematically evaluate the findings.

We briefly also considered combining all outcome variables of the same probability distribution (Gaussian, binomial) into a single model, with a random effect for outcome type (Depression, BMI, etc.); this would allow us to estimate a single overall effect of each predictor, with random slopes for each outcome type. However, we decided against this approach because it hinges on some strong assumptions, such as the residuals of all Gaussian outcome variables having the same dispersion captured by a single σ parameter. Instead, readers can use the coefficient plots and Cohen’s d values to evaluate effect sizes on a case by case basis. In combination with the probabilistic inference facilitated by Bayesian modeling we believe this provides readers with all the necessary information to draw their own conclusions.

We have also added a more detailed treatment of causal inference (see also response to next comment) by adding DAGs and additional text in Appendix 1.

Finally, the first paragraph of the discussion provides a summary of the findings in relation to the predictions.

4. This may vary by discipline, but it is inappropriate to test mediators by adjusting for them as covariates in epidemiology. Mediation is normally assessed using causal modelling techniques. Furthermore, the study appears to use a cross-sectional data set making it difficult to distinguish between cause and effect irrespective of methods used. Therefore, the study cannot provide any definitive mediational results. The Introduction nicely summarises the theoretical basis of a psychosocial stress pathway, so perhaps the authors could explore the potential role of psychosocial factors in wealth-health associations without making strong claims. By reporting that there was weak evidence of associations between wealth/wealth inequality with psychosocial factors (paths A) and between psychosocial factors and health outcomes (paths B), in turn, according to the Baron and Kenny approach, one could say that there was weak, albeit indirect, evidence of these factors as potential mediators. Given the hypothesised relations, it is clearer not to refer to psychosocial factors when referring to health outcomes and vice versa.

We agree that our approach to the mediation analysis was a bit too informal. We have addressed this in several ways. First, as mentioned above, we created a separate section of the results for reporting the mediation analyses, with the subheading “Do psychosocial variables mediate relationships between wealth or wealth inequality and health?”. Second, we begin this section by outlining the formal criteria for mediation according to Baron and Kenny (Lines 467-479):

“We tested the prediction (P3) that the effects of wealth or inequality on health were mediated via psychosocial pathways using formal mediation analysis (Baron and Kenny, 1986; MacKinnon et al., 2007). […] See Appendix 1 for a discussion and graphical depiction of the causal relationships assumed by this mediation approach.”

Third, we more clearly highlight cases where path A and path B are well supported, before mentioning that indirect effects were largely negligible, and that there was little evidence overall for mediation:

“The only convincing evidence for mediation was found for depression and non-social problems mediating the effect of household wealth on diastolic blood pressure; specifically, household wealth was negatively associated with diastolic blood pressure (path c) as well as with depression and non-social problems (paths a; see above, Figure 2), and both depression (β=-0.03, P>0=0.20) and non-social problems (β=0.08, P>0=0.05) were themselves negatively associated with diastolic blood pressure (paths b). However, there were no other cases where both path a and path b were well supported, the indirect effects of household wealth, community wealth, or inequality were virtually always 0 for any mediator (including depression and non-social problems), and the proportion mediated was generally small or highly uncertain (Tables S18-S20). Overall, there was little evidence of mediation.”

Finally, we added Appendix 1 which discusses the mediation analysis and its causal assumptions in more detail. Specifically, we present formal DAGs of the causal relationships assumed by the mediation analyses we performed and highlight potential problems due to treating each mediator independently, when in fact two or more mediators might be causally affected by wealth/inequality and/or each other. As discussed in detail in Appendix 1, we ultimately don’t think this is a problem because our previous analysis included several mediators simultaneously and arrived at the same conclusion as the present approach: that there was little evidence for wealth/inequality associations with health to be mediated by the “psychosocial” variables we used.

All that being said, the reviewers are right that the data are largely cross-sectional and hence we cannot provide causality in the Granger sense (e.g. increases in wealth inequality precede increases in psychosocial stress which then lead to detrimental health outcomes). We now also mention this explicitly in the Discussion (Lines 663-664):

“a limitation of our data is that we cannot determine the direction of causation given our cross-sectional design”.

5. The results for community-level inequality measures were difficult to understand. Based on Figure 1, it looks like higher inequality communities have small samples, with the exception of Mission. Do these sample size differences affect the reliability of the inequality effect estimates? Related: are community-level measures based on the households sampled, or on the entire community including individuals who may not be in some of the analyses? Or are household samples exhaustive for these communities? In the inequality analysis, do measures based only on the households sampled (if non-exhaustive) affect whether the Gini coefficients are representative of the community (especially in small samples, e.g. n=9)?

Thanks for these good questions and the opportunity to clarify. Originally, we had used number of households sampled as our measure of community size, rather than full community size. We replaced this variable with the actual number of adults in the community (based on a complete census). In practice this made little difference, since sample size per community and complete community size are highly correlated, meaning we sampled relatively evenly across communities (r = 0.95).

Gini coefficients are necessarily based just on the households with wealth measured, and not the complete community. However, they are not limited to individuals with other covariates, so remain the same across analyses. It is plausible that Gini measures might be less reliable for smaller samples, however, we find no evidence for systematic bias in Gini estimates by community sample size. We’ve added the following to the paper to further characterize the relationships between community level variables (lines 311-317):

“Wealth inequality was generally higher in communities closer to the market towns of San Borja and Yucumo, where Tsimane can sell produce and purchase market goods, though some villages near towns show low inequality (Figure 1D) (correlation between Gini and distance to market r=-0.38, df=38, p=0.01). Inequality was marginally lower in richer communities (r=0.22, df=38, p=0.17). Community size was not significantly related to distance (r=-0.18, p=0.26), mean wealth (r=0.11, p=0.50), or inequality (r=0.00, p=0.99). ”

6. We found it challenging to interpret some of the results without more detail on the modelling strategy. Even after reading the methods, it was unclear who was included in the analysis. For example, in Table 1 the age range is 0-89.7 years. Does this mean that infants, children, and adolescents are grouped in the analysis with adults? If so, what's the justification for this approach, given that these age groups may experience wealth, morbidity, and mortality risk in qualitatively different manners (not captured by modeling age as a covariate)? Page 8 claims that the study includes 1/3 of the Tsimane population so does the analytic sample include approx. 5,300 persons of the 16K?

Thanks for pointing this out. We’ve updated table 1 to make the sample clearer, reporting the number of unique individuals and number of observations for each variable, and further clarifying sample size at the individual, household, and community levels. Additionally, each model in the supplementary tables now lists the sample size for that analysis in terms of observations and clustering at all three levels.

In light of the comment about the wide age range, we have now divided the sample and analyses into adults > 15 years of age (since the youngest head of household in our sample is 16) and juveniles =< 15. This makes clear that some variables were collected only on adults, especially the psychosocial outcomes, and also allows us to show where effects are similar or different for juveniles vs. adults. There are now separate tables, figures, and results for adults and juveniles.

To further clarify who was included in the analysis we added a figure supplement to Figure 1, with the caption:

“Ever sampled by THLHP refers to the period potentially included in this study, i.e. up to December 2015; note that this sample includes 92 communities. […] For a further missingness breakdown of the sample by specific outcome variable see Table 1”.

We hope that this helps clarify the sample composition relative to the larger Tsimane population. We also corrected the statement mentioned in the comment to say that the sample included “*approximately one quarter of the Tsimane population” (Line* 213).

7. For the morbidity data, were morbidities counted based on occurrence at the time of the medical check-up, or also retrospectively across some span of time? If the former, to what degree do you think this would introduce measurement error?

These are based on occurrence at the time of medical check-up, with repeat check-ups for some individuals. We now make this more explicit in lines 812-813:

“Morbidity at the time of the medical check-up was assessed by physicians using the ICD-10 classification”.

Likely this is less sensitive than a continuous measure over time would be. However, since prevalence’s are high (22-36%) and many infections are chronic (especially parasitic ones), we don’t think this is a major limitation.

8. Since Bayesian imputation might lead to some errors in estimating effects, the authors should comment on this. Furthermore, statistical methods to deal with missing data are useful to a degree, but Methods should report sample sizes and the proportion and reason for such high levels of missing data. If we read Table 1 correctly, the number of adults with health data vary from 850 adults for self-rated health to 9000 for BMI? Data on psychosocial factors are also quite low – if missing data are so high, how reliable are these data? Is this why there was weak evidence on the role of psychosocial factors? The study should provide information on the data quality in Methods and reflect on these limitations in Discussion.

In the original version imputation was only used in the mediation models since in those models we needed to combine samples on multiple different outcomes that only partially overlapped. We have since redone those models (see response to comment 4) and have decided not to use imputation, given the large amount of missing data from some overlapping samples. The actual sample size for these models is now noted, and we have added a note about sample limitations for mediation analyses in the discussion (line 684-686):

“An obvious limitation is that our sample sizes for the mediation analysis were smaller than for other analyses (Table S18-S20), though most were still large enough to capture any meaningful effect.”

More generally, however, the reason some variables have much higher missing values is because the larger epidemiological surveillance of our project always collects basic information like BMI and medical diagnoses, but hormone sampling, depression, etc. were separate intersecting studies on sub-samples of our larger population – most done either with stratified age sampling, or on ages 40+ (given the primary THLHP research focus on aging).

9. There was some discussion between reviewers as they had some differences in interpretation of the results. One reviewer found that wealth inequality-health effects were more substantial than wealth-health, which is contrary to the authors' interpretation. For example, if comparing Figures 2 and 3, wealth associations with health were mixed across outcomes (3 strong associations out of 8), but wealth inequality associations were more consistent (5 strong associations out of 8). If this interpretation is correct, it would be important to consider the relative inconsistency/consistency of associations between outcomes since the paper sometimes refer to key findings for "health" as a global metric. Among the paper's strengths is its inclusion of a wide range of unique health outcomes, making it important to highlight any differences in results between health outcomes and to discuss potential reasons why household wealth is a weaker predictor of systolic and diastolic blood pressure than wealth inequality, for example.

Firstly, as mentioned above, the new figures 2-5 (coefficient plots), the more consistent coding of outcome variables, and the standardized effect sizes should make it easier to evaluate the relative importance of predictor variables. Secondly, we’ve strengthened the discussion quite a bit in terms of the implications of findings, particularly on blood pressure and respiratory illness, which are probably the two most meaningful associations. In editing the discussion we’ve mostly avoided general discussions about “health” as a global metric, because overall the results for both wealth and inequality are quite mixed. We also don’t discuss the number of associations as a metric, since to some extent this is artificial (i.e. there are two measures for blood pressure, total morbidity incorporates the binary diagnoses that are presented separately, etc.) – this is another reason why we avoided analyzing all outcomes in a single multivariate model to generate an “overall effect” of wealth / inequality (see response to comment 3). Beyond highlighting that greater wealth in general was more consistently associated with beneficial outcomes than lower inequality (1^st^ paragraph of discussion) and discussing the strongest associations, i.e. respiratory disease and blood pressure, and their potential mechanisms (2^nd^ and 3^rd^ paragraph of the discussion, respectively), as well as links between psychosocial factors and health (6^th^ paragraph) we found that comparing other associations (or lack thereof) in more detail would be too speculative and take up too much space. However, the results are now presented in a way that should more easily facilitate readers’ evaluation of specific findings.

10. The limitation of the difficulty of disentangling association vs. causation and the potential of reverse causation (due to the cross-sectional design) needs to be addressed more directly and clearly. If inferential analyses cannot be made to support the causation case, the quality of the evidence being presented in the paper should be discussed more critically.

The reviewers are correct that we cannot disentangle association from causation given our study design. We’ve now added the following to the discussion to make these limitations more explicit (lines 656-665):

“Another possible interpretation for some of these associations may be that causality is reversed, with poor health leading to less wealth or exacerbated inequality. […] Other confounds might also be possible, for example if people preferentially assort by health or wealth by moving between villages.”

11. The paper is written in an evolutionary medicine framework, with contrasts drawn between a "trade-offs" model and a mismatch model. This is an interesting presentation, but I'm concerned that the mismatch model is essentially unfalsifiable here. If social gradients did not exist in the Tsimane, mismatch would probably be invoked. But based on the results presented here, they do, including for markers of cardiovascular health-and this is also presented as support for evolutionary mismatch. An equally likely interpretation is that responses to relative wealth and social inequality across different types of human societies are shared, counter to the mismatch hypothesis. Because mismatch can be supported whether or not health-wealth associations are found, P4a and P4b become somewhat poorly defined predictions. For P4b, it's also worth noting that social gradients in health are observed in high-income countries for both chronic and infectious disease (the current pandemic is a good example of the latter). The results are valuable to understanding the evolutionary history of social gradients in humans, but a more critical take on which hypotheses it can robustly test would improve the manuscript.

These are very good points and we agree that the predictions related to mismatch were underspecified. We have now refrained from attempting to make contrasting predictions regarding mismatch vs tradeoffs. Instead, we focus solely on the more fundamental predictions that relatively higher socio-economic position should have beneficial effects (P1), that higher inequality should have detrimental effects (P2), and that associations between wealth/inequality and health should be mediated by psychosocial stress (P3); these are the predictions that follow from the “relative wealth hypothesis”, the “inequality hypothesis” and their proposed biological mechanisms (see e.g. Lynch et al. 2004).

We have kept the evolutionary medicine framing though and expanded the introduction of the mismatch concept in lines 121-135. Rather than presenting tradeoffs and mismatch as contrasting hypotheses, we now frame them as complementary: people might have evolved to shift their behavior and physiology in particular ways depending on local context, e.g. they might be more aggressive, have higher cortisol levels and higher blood pressure when living in a highly competitive environment, and these shifts might become especially detrimental to health when coupled with evolutionarily novel lifestyles. For instance, environments may be more competitive than before, leading to even higher levels of cortisol or blood pressure, and/or this altered physiology is especially damaging in combination with novel risk factors like obesogenic diets, lack of physical activity, etc., leading to chronic disease. In other words, what’s novel in a mismatched environment is not that physiology responds to the competitive environment, but that this response is more likely to cause disease. We hope that this expanded introduction makes it easier to understand our interpretation and discussion of the blood pressure results (lines 622-653): that we see here the beginnings of what seems to be such a common problem in modern societies – that blood pressure is affected by the socioeconomic context – and while this may not cause much harm among the Tsimane, given their overall excellent cardiovascular health, it lays the foundation for chronic problems when coupled with novel risk factors.

[Editors' note: further revisions were suggested prior to acceptance, as described below.]

The authors have addressed all points raised during the review. We commend the authors for their substantial efforts on the manuscript, including the additional analyses. We have only one further essential revision, which we hope will be a straightforward addition to the manuscript:1. In order to help readers understand the range of results (e.g., many observed associations were weak or uncertain), we recommend that the authors expand the discussion with a few statements about the high degree of heterogeneity in their findings explicitly, especially given the wealth of evidence from high-income developed nations which generally find strong and directionally consistent effects of SES on health.

We thank the editors and reviewers for their time and the positive evaluations, and for the opportunity to revise our paper. We have restructured the Discussion and added new writing, which we believe strengthens the Discussion substantially. Specifically, the first paragraph states all the main findings, including the heterogeneity of results (Lines 427 – 431):

“Overall, our results showed substantial heterogeneity in terms of the direction and magnitude of associations between wealth, wealth-inequality, and health, which contrasts with the more consistent SES-health gradients in high-income countries. Nevertheless, some findings supported an association between wealth or inequality and health outcomes, though these associations were not mediated by psychosocial factors.”

This is followed by the existing paragraphs discussing the clinical relevance of the respiratory disease and blood pressure results, as well as alternative explanations for these. Then follow several paragraphs putting the heterogeneity of results into context, all but the last one of which are new (Lines 507-553):

“Beyond respiratory disease and blood pressure, many associations were inconclusive. […] Thus, household wealth may in part be inconsistently associated with health because of these opposing processes.”Reviewer #3:[…] The results are still quite complex to work through, and many of the observed associations are weak or uncertain enough that it's unclear whether they're "real" (this is exacerbated somewhat by the many statistical tests, which raises questions about multiple hypothesis testing). To be clear, I do not think further or altered analyses are needed: the complexity in the results seems likely to reflect, at least in large part, real-world complexity in wealth-health relationships in this society. However, the authors might want to consider discussing the high degree of heterogeneity in their findings explicitly, especially considering the sense one gets from the literature on wealthy, developed nations, that SES effects on morbidity/mortality are often both strong and directionally consistent.

Thank you for the positive evaluation and for this suggestion. We’ve modified the discussion to better reflect this point, adding several paragraphs (see excerpts above). Specifically, we discuss the possibilities that wealth-health associations may be less consistent among the Tsimane because (i) the epidemiological context is dominated by infectious, rather than chronic disease, (ii) we did not assess wealth and inequality at larger scales, such as in comparison to the local rural population, or all of Bolivia, and (iii) household wealth reflects both local status as well as market integration, which have partly opposing effects on stress and possibly infectious disease exposure and thereby health.